# An integrative approach reveals five new species of highland papayas (Caricaceae, *Vasconcellea*) from northern Peru

Daniel Tineo[☯], Danilo E. Bustamante[ID]*[☯], Martha S. Calderon[☯], Jani E. Mendoza, Eyner Huaman, Manuel Oliva

Instituto de Investigación para el Desarrollo Sustentable de Ceja de Selva (INDES-CES), Universidad Nacional Toribio Rodríguez de Mendoza, Chachapoyas, Amazonas, Peru

☯ These authors contributed equally to this work.
* danilo.bustamante@untrm.edu.pe

**Data Availability Statement:** Materials are deposited at Herbarium Universidad Nacional Toribio Rodriguez de Mendoza (http://sweetgum. nybg.org/science/ih/herbarium-details/?irn=

## Abstract

The assignment of accurate species names is crucial, especially for those with confirmed agronomic potential such as highland papayas. The use of additional methodologies and data sets is recommended to establish well-supported boundaries among species of *Vasconcellea*. Accordingly, six chloroplast (*trnL-trnF*, *rpl20-rps12*, *psbA-trnH* intergenic spacers, *matK* and *rbcL* genes) and nuclear (ITS) markers were used to delimit species in the genus *Vasconcellea* using phylogeny and four DNA-based methods. Our results demonstrated congruence among different methodologies applied in this integrative study (i.e., morphology, multilocus phylogeny, genetic distance, coalescence methods). Genetic distance (ABGD, SPN), a coalescence method (BPP), and the multilocus phylogeny supported 22–25 different species of *Vasconcellea*, including the following five new species from northern Peru: *V. badilloi* sp. nov., *V. carvalhoae* sp. nov., *V. chachapoyensis* sp. nov., *V. pentalobis* sp. nov., and *V. peruviensis* sp. nov. Genetic markers that gave better resolution for distinguishing species were ITS and *trnL-trnF*. Phylogenetic diversity and DNA-species delimitation methods could be used to discover taxa within traditionally defined species.

## Introduction

The family Caricaceae is composed of six genera containing 35 species that are distributed from southern Mexico to northern Chile [1, 2]. Two of these genera, namely *Carica* L. and *Horovitzia* Badillo are monospecific. The former is considered the most economically important and is distributed in tropical and subtropical America [3], whereas the latter is endemic to Mexico [4]. Three additional genera are *Cylicomorpha* Urban, *Jacaratia* A. DC, and *Jarilla* Rusby. The first one has two species and is the only genus restricted to the African premontane wet forestst [2]. The second one comprises seven species distributed along South America [2]. The latter one encompasses three species distributed along Pacific Coast from northern Mexico to El Salvador [5]. The remaining genus, namely *Vasconcellea* Saint-Hilaire, is the largest one in this family encompassing 20 species and 1 hibrid (*V.* x *heibornii*) distributed mainly from Ecuador to Peru [3, 6].

259051) which is indexed in the Index Herbariorum of the New York Botanical Garden. The voucher numbers: CHAX224-CHAX253. Images of these materials are included in the main manuscript. Additional information was included in https://knb. ecoinformatics.org under the following DOI doi:10. 5063/3X852P. All Genbank accession numbers are available from https://www.ncbi.nlm.nih.gov/ genbank/ under the following accession numbers: MT823582 - MT823725.

**Funding:** This research was funded by INDES-CES/ UNTRM throughout Project SNIP N˚ 312252 "FISIOVEG" and Project SNIP N˚ 352431 "PROCICEA". The funders had no role in study design, data collection and analysis, decision to publish, or preparation of the manuscript.

**Competing interests:** The authors have declared that no competing interests exist.

Initially, *Vasconcellea* was embedded into the genus *Carica* [7]; however, molecular analyses confirmed that these genera were not monophyletic [8, 9], and *Vasconcellea* was restored as a different genus [6]. The genus *Vasconcellea* is characterized by simple, lobed, or palm-lobed leaves with five to six main veins [10]. Besides, flowers have a corolla with curved lobules to the left, linear stigmas, five locule ovaries, and scattered ovules in two juxtaposed divisions [10]. The species richness of *Vasconcellea* shows that northern Andes are the areas with the highest diversity [1, 10]. *Vasconcellea* species are distributed from the dry slopes of the Andes in Ecuador, Colombia and Peru (3,500 m.a.s.l.) to the lowlands of Panama to southern Brazil and called highland papayas due to their climatic preferences [1]. Highland papayas have a number of desirable characteristics, such as disease resistance, cold tolerance, high latex enzymatic activity, and high protein and vitamin contents [10], which suggest their agronomic potential, especially in Andean towns [11]. Currently, eigth species of the genus *Vasconcellea* have been reported from northern (Amazonas, Cajamarca) to southern Peru (Moquegua): *V. candicans* (A.Gray) A. DC, *V. glandulosa* A. DC, *V. microcarpa* (Jacq.) A. DC, *V. monoica* (Desf.) A. DC, *V. parviflora* A. DC, *V. pubescens* A. DC, *V. quercifolia* A. St.-Hil., and *V. weberbaueri* (Harms) V.M. Badillo [2, 12–14].

The evolutionary history of these taxa might be misunderstood by recognizing distinct clades in single gene trees as species [15]. Therefore, the use of multilocus sequence data is crucial in the establishment of robust species boundaries [16, 17]. Several molecular-phylogenetic analyses of Caricaceae have been undertaken using isozymes, RFLP, and AFLP but none included representatives of all genera [8, 9, 14, 18–20]. Nuclear and plastid DNA sequences from all of the family's extant species was compiled [2] and the evolutionary relationships within the family Caricaceae have therefore clarified. These chloroplast (*trnL-trnF*, *rpl20-rps12*, *psbA-trnH* intergenic spacers, *matK* and *rbcL* genes) and nuclear sequences (ITS) have been recommended to assess inter- and intraspecific relationships among species of Caricaceae [2, 9]. Additionally, estimating species trees and establishing species boundaries among different taxa are challenging [15, 17]. This has been overcome by methods that encompass genetic distance and coalescent approaches, which have proven very useful and been widely used for a range of taxa [15, 17–20]. Accordingly, the use of several methodologies and data sets to delimit species (i.e., integrative approaches) is highly recommended, and subsequently, the achievement of congruent results across the methods is likely to prove most useful for framing reliably supported species boundaries [21–25].

In this study, we analysed species of the genus *Vasconcellea*, including new unreported taxa from northern Peru, based on an integrative approach (i.e., morphological observations, phylogenetic inferences, and DNA-species delimitation methods). Six molecular markers (ITS, *matK*, *psbA-trnH*, *rbcL*, *rpl20-rps12*, *trnL-trnF*) were used to examine the phylogenetic relationships and assess boundaries of species within the genus *Vasconcellea*.

## Materials and methods

### Collection of specimens

A total of 30 specimens of highland papayas were sampled from five provinces along the Region Amazonas, in northern Peru (Bongará, Chachapoyas, Luya, Rodriguez de Mendoza, Utcubamba; Fig 1). Permit of scientific research of wild flora (D000134-MINAGRI-SERFOR-DGGSPFFS-DGSPF) was provided by Servicio Nacional Forestal y de Fauna Silvestre (SERFOR). Tissue samples of approximately 50 mm$^2$ were taken from leaf tips for genetic analyses and placed in prelabelled 1.5 mL Safelock Eppendorf tubes. For each site, the date, time, and GPS coordinates were recorded. Photographs were taken to record sampling locations and site features. Samples were morphologically characterized according to Badillo [4, 6] and

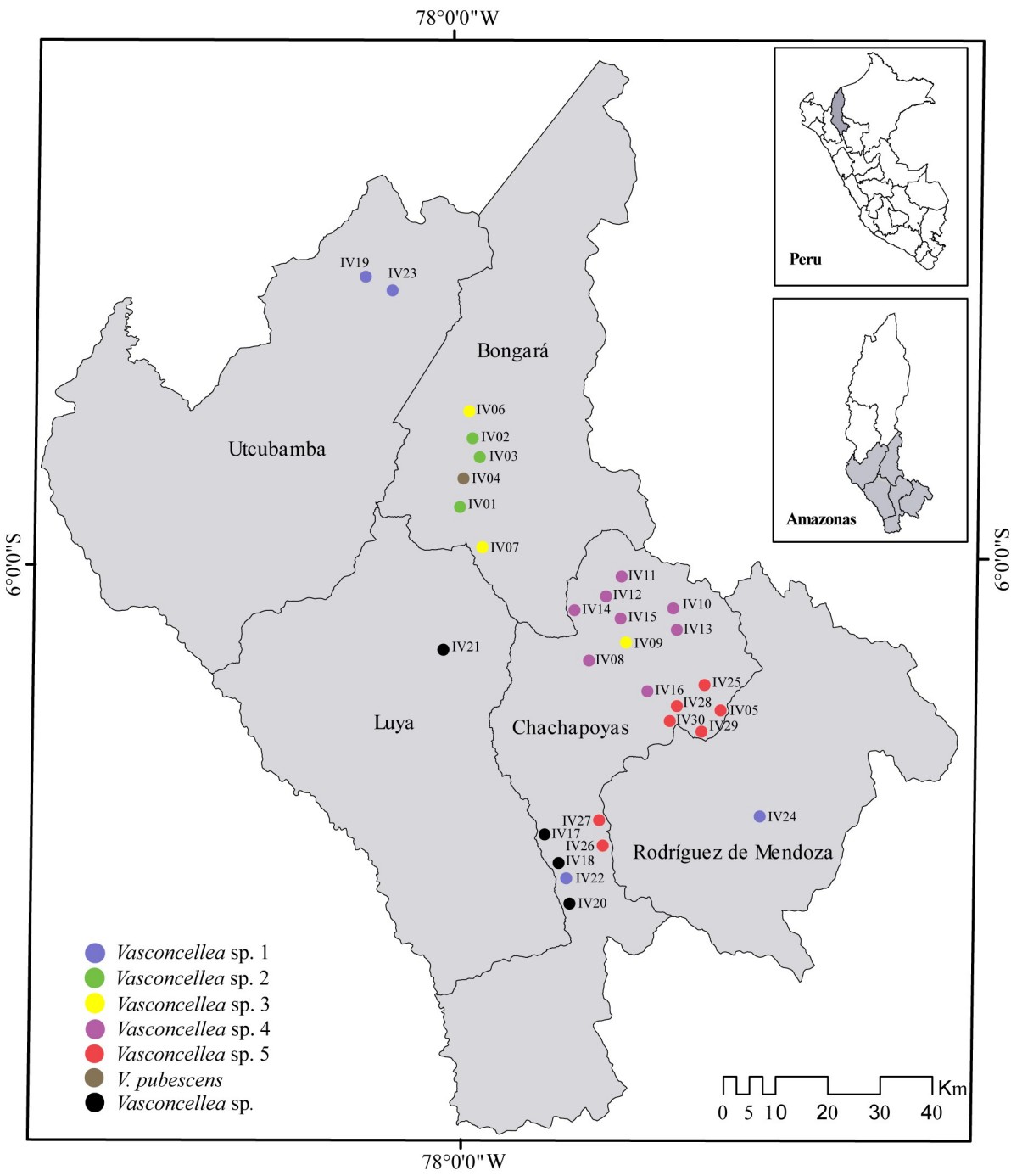

**Fig 1. Collections of the 30 specimens of the genus *Vasconcellea* from the Region Amazonas, northern Peru.** The national, provincial and district boundaries were obtained from the Geoportal of the National Geographic Institute of Peru (IGN) in shapefile format with a DATUM WGS 1984 for illustrative purposes only.

deposited in the herbarium of Universidad Nacional Toribio Rodríguez de Mendoza (CHAX), Peru (Table 1) [26]. Furthermore, records and morphology of *Vasconcellea* species were revised from databases and collections as JSTOR Global Plants (https://plants.jstor.org/), the New York Botanical Garden Steere herbarium (http://sweetgum.nybg.org/science/), the Global

**Table 1. List of samples of highland papayas collected in Region Amazonas, northern Peru.**

| Species | Code | Herbario Voucher | Place | Date | Elevation (m.a.s.l) | Latitude (South) | Longitude (West) |
|---|---|---|---|---|---|---|---|
| *V. badilloi* | IV06 | CHAX224 | Pomacochas, Bongará | 13 Sep. 2018 | 2280 | 5°48'53" | 77°57'24" |
| *V. badilloi* | IV07 | CHAX225 | Cuchulia, Bongará | 13 Sep. 2018 | 1386 | 5°59'44" | 77°58'30" |
| *V. badilloi* | IV09 | CHAX226 | Quinjalca, Chachapoyas | 20 Sep. 2018 | 3143 | 6°05'33" | 77°40'39" |
| *V. carvalhoae* | IV01 | CHAX227 | Pomacochas, Bongará | 13 Sep. 2018 | 2401 | 5°49'45" | 77°58'12" |
| *V. carvalhoae* | IV02 | CHAX228 | Pomacochas, Bongará | 13 Sep. 2018 | 2263 | 5°49'08.7" | 77°57'39.3" |
| *V. carvalhoae* | IV03 | CHAX229 | Pomacochas, Bongará | 13 Sep. 2018 | 2236 | 5°49'37" | 77°58'01" |
| *V. chachapoyensis* | IV08 | CHAX230 | Quinjalca, Chachapoyas | 20 Sep. 2018 | 3130 | 6°05'30.4" | 77°40'30.4" |
| *V. chachapoyensis* | IV10 | CHAX231 | Granada, Chachapoyas | 20 Sep. 2018 | 2996 | 6°06'12" | 77°37'47" |
| *V. chachapoyensis* | IV11 | CHAX232 | Olleros, Chachapoyas | 20 Sep. 2018 | 3041 | 6°03'07" | 77°38'54" |
| *V. chachapoyensis* | IV12 | CHAX233 | Olleros, Chachapoyas | 20 Sep. 2018 | 3031 | 6°03'13.2" | 77°38'47.3" |
| *V. chachapoyensis* | IV13 | CHAX234 | Granada, Chachapoyas | 20 Sep. 2018 | 3017 | 6°06'10" | 77°37'39" |
| *V. chachapoyensis* | IV14 | CHAX235 | Asunción, Chachapoyas | 20 Sep. 2018 | 2821 | 6°01'56.6" | 77°42'37.1" |
| *V. chachapoyensis* | IV15 | CHAX236 | Quinjalca, Chachapoyas | 20 Sep. 2018 | 3150 | 6°05'25" | 77°40'46" |
| *V. chachapoyensis* | IV16 | CHAX237 | San José, Chachapoyas | 20 Sep. 2018 | 2200 | 6°16'59.2" | 77°33'31.7" |
| *V. pentalobis* | IV05 | CHAX238 | Ocol, Chachapoyas | 05 Sep. 2018 | 2297 | 6°14'49" | 77°32'50" |
| *V. pentalobis* | IV25 | CHAX239 | Ocol, Chachapoyas | 08 Feb. 2019 | 2406 | 6°15'35" | 77°32'45" |
| *V. pentalobis* | IV26 | CHAX240 | Cuchapata, Chachapoyas | 08 Feb. 2019 | 2523 | 6°28'25" | 77°42'13" |
| *V. pentalobis* | IV27 | CHAX241 | Cuchapata, Chachapoyas | 15 Feb. 2019 | 2342 | 6°28'25" | 77°41'51" |
| *V. pentalobis* | IV28 | CHAX242 | Izcuchaca, Chachapoyas | 15 Feb. 2019 | 2385 | 6°20'08" | 77°31'49" |
| *V. pentalobis* | IV29 | CHAX243 | Izcuchaca, Chachapoyas | 15 Feb. 2019 | 2321 | 6°20'41" | 77°31'17" |
| *V. pentalobis* | IV30 | CHAX244 | Izcuchaca, Chachapoyas | 15 Feb. 2019 | 2659 | 6°18'59" | 77°33'21" |
| *V. peruviensis* | IV19 | CHAX245 | Buenos Aires, Utcubamba | 12 Aug. 2019 | 1571 | 5°40'33.1" | 78°20'23.8" |
| *V. peruviensis* | IV22 | CHAX246 | Cueyqueta, Chachapoyas | 16 Oct. 2018 | 2557 | 6°31'33.0" | 77°48'50.2" |
| *V. peruviensis* | IV23 | CHAX247 | Buenos Aires, Utcubamba | 12 Aug. 2018 | 1538 | 5°40'04" | 78°20'17" |
| *V. peruviensis* | IV24 | CHAX248 | Santa Rosa, Rodríguez de Mendoza | 16 Aug. 2018 | 1887 | 6°26'35.4" | 77°28'44.9" |
| *V. pubescens* | IV04 | CHAX249 | Pomacochas, Bongará | 12 Sep. 2019 | 2285 | 5°48'31" | 77°57'09" |
| *V. stipulata* | IV17 | CHAX250 | Péngote, Chachapoyas | 26 Oct. 2018 | 2523 | 6°32'58.1" | 77°48'52.4" |
| *V. stipulata* | IV18 | CHAX251 | Péngote, Chachapoyas | 26 Oct. 2018 | 2422 | 6°32'23" | 77°48'56" |
| *V. stipulata* | IV20 | CHAX252 | Cueyqueta, Chachapoyas | 26 Oct. 2018 | 2557 | 6°31'33.0" | 77°48'50.2" |
| *V. stipulata* | IV21 | CHAX253 | Lamud, Luya | 30 Oct. 2018 | 2311 | 6°08'22.9" | 77°57'03.1" |

Biodiversity Information Facility (https://www.gbif.org/), and Tropicos from Missouri Botanical Garden (http://www.tropicos.org).

## DNA sequencing and alignment preparation

Genomic DNA was extracted from leaf tissue using the NucleoSpin Plant II Kit (Macherey-Nagel, Düren, Germany) following the manufacturer's instructions. Six molecular markers were sequenced (ITS, *matK*, *psbA-trnH*, *rbcL*, *rpl20-rps12*, *trnL-trnF*). Each gene was amplified using polymerase chain reaction (PCR) with MasterMix (Promega, Wisconsin, USA) in the following reaction mixture: 10 ng of DNA and 0.25–0.5 pmol of forward and reverse primers for a total volume of 10 µl. The PCR protocols followed Bustamante et al. [17, 27], and primer combinations are summarized in S1 Table. The sequences of the forward and reverse strands were determined commercially by Macrogen Inc. (Macrogen, Seoul, Korea). New generated sequences from the six markers were deposited in GenBank. These sequences and others obtained from GenBank (Table 2) were initially aligned with Muscle algorithms [28] and were adjusted manually with MEGA6 software [29].

**Table 2. List of species used in the molecular analyses.**

| Specie | Country | Voucher | ITS | *mat*K | *psb*A-*trn*H | *rbc*L | *rpl*20-*rps*12 | *trn*L-*trn*F |
|---|---|---|---|---|---|---|---|---|
| *Carica papaya* | Guatemala | KJ399 | AY461564 | JX092002 | JX091963 | JX091913 | JX091875 | JX091823 |
| | Ecuador | RPEH57 | JX092051 | JX092003 | AY847053 | JX091914 | JX091874 | DQ061124 |
| *Cylicomorpha parviflora* | Tanzania | MMA3212 | JX092052 | JX092004 | JX091964 | JX091915 | JX091876 | JX091824 |
| *Cylicomorpha solmsii* | Cameroon | GJP2115 | JX092053 | JX092005 | JX091965 | JX091916 | JX091877 | JX091825 |
| *Horovitzia cnidoscoloides* | Mexico | TRC8167 | JX092054 | JX092006 | JX091966 | JX091917 | JX091878 | JX091826 |
| *Jacaratia corumbensis* | Paraguay | FK1468 | JX092056 | JX092008 | JX091969 | JX091918 | JX091879 | JX091829 |
| *Jacaratia dolichaula* | Mexico | CJI4785 | JX092058 | JX092010 | JX091970 | JX091920 | JX091881 | JX091832 |
| *Jacaratia spinosa* | Peru | HE1348 | JX092062 | JX092015 | JX091972 | JX091925 | JX091883 | JX091836 |
| *Jacaratia* sp. | Peru | HE1365 | JX092063 | JX092013 | JX091974 | JX091923 | JX091882 | JX091827 |
| *Jarilla caudata* | Mexico | LJA20002 | JX092065 | JX092016 | JX091975 | JX091926 | JX091885 | JX091839 |
| *Jarilla chocola* | Mexico | LEJ31 | JX092064 | JX092017 | JX091977 | JX091927 | JX091884 | JX091838 |
| *Jarilla heterophylla* | Mexico | LJA20002 | JX092066 | JX092018 | JX091978 | JX091928 | JX091886 | JX091840 |
| **V. badilloi** | Peru | IV06 | x | MT823587 | MT823611 | MT823641 | MT823671 | MT823701 |
| | Peru | IV07 | x | MT823588 | MT823612 | MT823642 | MT823672 | MT823702 |
| | Peru | IV09 | x | MT823590 | MT823614 | MT823644 | MT823674 | MT823704 |
| *V. candicans* | Peru | SL1201 | JX092074 | JX092025 | JX091986 | JX091936 | JX091892 | JX091848 |
| **V. carvalhoae** | Peru | IV01 | x | MT823582 | MT823606 | MT823636 | MT823666 | MT823696 |
| | Peru | IV02 | MT808984 | MT823583 | MT823607 | MT823637 | MT823667 | MT823697 |
| | Peru | IV03 | x | MT823584 | MT823608 | MT823638 | MT823668 | MT823698 |
| *V. cauliflora* | Guatemala | SPC89272 | JX092075 | JX092028 | JX091987 | JX091939 | JX091894 | JX091850 |
| **V. chachapoyensis** | Peru | IV08 | MT808987 | MT823589 | MT823613 | MT823643 | MT823673 | MT823703 |
| | Peru | IV10 | MT808988 | MT823591 | MT823615 | MT823645 | MT823675 | MT823705 |
| | Peru | IV11 | MT808989 | MT823592 | MT823616 | MT823646 | MT823676 | MT823706 |
| | Peru | IV12 | MT808990 | MT823593 | MT823617 | MT823647 | MT823677 | MT823707 |
| | Peru | IV13 | MT808991 | MT823594 | MT823618 | MT823648 | MT823678 | MT823708 |
| | Peru | IV14 | x | MT823595 | MT823619 | MT823649 | MT823679 | MT823709 |
| | Peru | IV15 | MT808992 | MT823596 | MT823620 | MT823650 | MT823680 | MT823710 |
| | Peru | IV16 | MT808993 | MT823597 | MT823621 | MT823651 | MT823681 | MT823711 |
| *V. chilensis* | Chile | FCsn | JX092076 | JX092030 | JX091990 | JX091941 | JX091895 | JX091852 |
| *V. crassipetala* | Ecuador | RPEH282 | AY461530 | AY461559 | AY847039 | JX091942 | JX091896 | DQ061132 |
| *V. glandulosa* | Argentina | NLJ8655 | JX092077 | JX092033 | JX091991 | JX091943 | JX091897 | JX091854 |
| *V. goudotiana* | Colombia | RPEH285 | AY461540 | JX092035 | AY847035 | JX091945 | JX091899 | DQ061135 |
| *V. x heilbornii* | Ecuador | RPEH155 | AY461528 | JX092037 | x | JX091947 | x | DQ061127 |
| *V. horovitziana* | Ecuador | RM262683 | AY461543 | AY461566 | AY847036 | x | x | DQ061141 |
| *V. longiflora* | Ecuador | RPEH228 | AY461542 | AY461557 | AY847037 | x | x | DQ061131 |
| *V. microcarpa* | Ecuador | RPEH225 | AY461536 | AY461563 | AY847052 | JX091948 | x | DQ061130 |
| *V. monoica* | Ecuador | RPEH58 | AY461537 | JX092039 | AY847032 | JX091950 | JX091901 | DQ061119 |
| *V. omnilingua* | Ecuador | RPEH238 | AY461534 | JX092040 | AY847042 | JX091951 | JX091902 | DQ061120 |
| *V. palandensis* | Ecuador | RPEH66 | AY461535 | JX092041 | AY847047 | JX091952 | JX091903 | DQ061140 |
| *V. parviflora* | Ecuador | RPEH45 | AY461526 | JX092043 | AY847048 | JX091954 | JX091905 | DQ061122 |
| **V. pentalobis** | Peru | IV05 | MT808986 | MT823586 | MT823610 | MT823640 | MT823670 | MT823700 |
| | Peru | IV25 | MT808999 | x | MT823630 | MT823660 | MT823690 | MT823720 |
| | Peru | IV26 | MT809000 | x | MT823631 | MT823661 | MT823691 | MT823721 |
| | Peru | IV27 | MT809001 | x | MT823632 | MT823662 | MT823692 | MT823722 |
| | Peru | IV28 | MT809002 | x | MT823633 | MT823663 | MT823693 | MT823723 |
| | Peru | IV29 | MT809003 | x | MT823634 | MT823664 | MT823694 | MT823724 |
| | Peru | IV30 | MT809004 | x | MT823635 | MT823665 | MT823695 | MT823725 |

(*Continued*)

**Table 2.** (Continued)

| Specie | Country | Voucher | ITS | *mat*K | *psb*A-*trn*H | *rbc*L | *rpl*20-*rps*12 | *trn*L-*trn*F |
|---|---|---|---|---|---|---|---|---|
| *V. peruviensis* | Peru | IV19 | **MT808994** | **MT823600** | **MT823624** | **MT823654** | **MT823684** | **MT823714** |
| | Peru | IV22 | **MT808996** | **MT823603** | **MT823627** | **MT823657** | **MT823687** | **MT823717** |
| | Peru | IV23 | **MT808997** | **MT823604** | **MT823628** | **MT823658** | **MT823688** | **MT823718** |
| | Peru | IV24 | **MT808998** | **MT823605** | **MT823629** | **MT823659** | **MT823689** | **MT823719** |
| *V. pubescens* | Peru | FHsn | JX092082 | JX092044 | KU664502 | JX091955 | JX091906 | JX091865 |
| | Peru | IV04 | **MT808985** | **MT823585** | **MT823609** | **MT823639** | **MT823669** | **MT823699** |
| *V. pulchra* | Ecuador | RPEH191 | AY461541 | AY461557 | AY847046 | x | x | DQ061128 |
| *V. quercifolia* | Bolivia | FTsn | JX092083 | JX092046 | JX091998 | JX091957 | JX091909 | JX091868 |
| *V. sphaerocarpa* | Colombia | SP6786 | JX092079 | JX092048 | JX091993 | JX091946 | JX091911 | JX091871 |
| *V. sprucei* | Ecuador | AE8784 | JX092085 | x | JX092001 | JX091960 | - | JX091872 |
| *V. stipulata* | Ecuador | RPEH55 | AY461548 | JX092049 | AY847051 | JX091961 | JX091912 | DQ061123 |
| | Peru | IV17 | x | **MT823598** | **MT823622** | **MT823652** | **MT823682** | **MT823712** |
| *V. weberbaueri* | Ecuador | RPEH10 | AY461527 | AY461573 | x | JX091962 | x | DQ061121 |
| ***Vasconcellea* sp.** | Peru | IV18 | x | **MT823599** | **MT823623** | **MT823653** | **MT823683** | **MT823713** |
| | Peru | IV20 | x | **MT823601** | **MT823625** | **MT823655** | **MT823685** | **MT823715** |
| | Peru | IV21 | **MT808995** | **MT823602** | **MT823626** | **MT823656** | **MT823686** | **MT823716** |
| *Moringa oleifera* (Outgroups) | India | CFA2227 | JX092069 | KY697380 | JX091981 | JX091931 | JX091889 | DQ061137 |
| *Moringa hildebrandtii* (Outgroups) | Madagascar | CFA2228 | JX092068 | JX092020 | JX091980 | JX091930 | JX091888 | JX091842 |

## Phylogenetic analyses

The phylogeny was based on concatenated data of the six molecular markers (65 sequences, Table 2). Selection of the best-fitting nucleotide substitution model was conducted using the program PartitionFinder [30] with six partitions (ITS, *matK*, *psbA-trnH*, *rbcL*, *rpl20-rps12*, *trnL-trnF*). The best partition strategy and model of sequence evolution were selected based on the Bayesian Information Criterion (BIC). The general time reversible nucleotide substitution model with a gamma distribution and a proportion of invariable sites (GTR + Γ + I) was selected for all partitions. Maximum likelihood (ML) analyses were conducted with the RAxML HPC-AVX program [31] implemented in the raxmlGUI 1.3.1 interface [32] using a GTRGAMMAI model with 1000 bootstrap replications. Bayesian inference (BI) was performed with MrBayes v. 3.2.6 software [33] using Metropolis-coupled MCMC and the GTR + Γ + I model. We conducted two runs each with four chains (three hot and one cold) for 10,000,000 generations, sampling trees every 1,000 generations. We plotted likelihood vs. generation using the Tracer Version v. 1.6 program [34] to reach a likelihood plateau and set the burn-in value.

## DNA-based species delimitation

We explored four different DNA-based delimitation methods using ITS, *matK*, *psbA-trnH*, *rbcL*, *rpl20-rps12*, and *trnL-trnF* data sets to assess species boundaries in *Vasconcellea*. Two of these DNA-based delimitation methods are based on genetic distance (statistical parsimony network analysis (SPN) [35] and automatic barcoding gap detection (ABGD) [36]) and two are based on coalescence (generalized mixed Yule coalescent method (GMYC) [37] and Bayesian phylogenetics and phylogeography (BPP) [38]).

For the SPN analyses of six markers, data sets were generated in TCS 1.21 [39] with a maximum connection probability set at 95% statistical confidence. The ABGD method was tested via a web interface (ABGD web, http://www.abi.snv.jussieu.fr/public/abgd/abgdweb.html).

Before analysis, the model criteria were set as follows: variability (P) between 0.001 (Pmin) and 0.1 (Pmax), minimum gap width (X) of 1.0, Kimura-2-parameters and 50 screening steps.

To perform the GMYC delimitation method, an ultrametric tree was constructed in BEAST v.2.0.2 [40], relying on the uncorrelated lognormal relaxed clock, the GTR + Γ + I model, and a prior coalescent tree. Bayesian Markov chain Monte Carlo simulation was run for 50 million generations, and trees and parameters were sampled every 1000 generations. Log files were visualized in Tracer v.1.6 [34] for assessing the stationary state of parameters on the basis of the value of estimate-effective sample size (ESS). After removing 25% of trees as burn-in, the remaining trees were used to generate a single summarized tree in TreeAnnotator v.2.0.2 [40] as an input file for GMYC analyses. The GMYC analyses with a single threshold model were performed in R (R Development Core Team, http://www.R-project.org) under the 'splits' package using the 'gmyc' function (R-Forge, http://r-forge.r-project.org/projects/splits/).

To validate the outcomes of single locus species delimitation, a multilocus BPP was applied using the program BP&P v.2.0 [41, 42]. The six markers were used as inputs for BPP under the A11 model (A11: species delimitation = 1, species tree = 1). Specimens were *a priori* assigned to species based only on the minimum number of species from the results of the phylogenetic analysis. Five variables (ε1~ε5) were automatically fine-tuned following the instructions of BP&P [41]. The prior distribution of θ could have influenced the posterior probabilities for different models [41]. Analyses were run with three different prior combinations [43]. Each analysis was run five times to confirm consistency between runs. Two independent MCMC analyses were run for 100,000 generations with the 'burn-in' = 20,000.

### Nomenclature

The electronic version of this article in Portable Document Format (PDF) in a work with an ISSN or ISBN will represent a published work according to the International Code of Nomenclature for algae, fungi, and plants, and hence the new names contained in the electronic publication of a PLOS article are effectively published under that Code from the electronic edition alone, so there is no longer any need to provide printed copies.

In addition, new names contained in this work have been submitted to IPNI, from where they will be made available to the Global Names Index. The IPNI LSIDs can be resolved and the associated information viewed through any standard web browser by appending the LSID contained in this publication to the prefix http://ipni.org/. The online version of this work is archived and available from the following digital repositories: PubMed Central, LOCKSS.

## Results

### Molecular phylogeny

In the phylogeny of *Vasconcellea* species, the analysed data matrix included a total of 5208 base pairs (bp) (723 bp for ITS, 1089 bp for *matK*, 736 bp for *psb*A-*trn*H, 1362 bp for *rbcL*, 802 bp for *rpl*20-*prs*12, and 496 bp for *trn*L-*trn*F) from 65 individuals. Phylogenetic trees obtained from the ML and BI analyses confirmed the monophyly of the genus *Vasconcellea* and its sister relationship to the genus *Jacaratia* (Fig 2). Our multilocus phylogeny also molecularly confirmed 19 of the 21 recognized species of *Vasconcellea*, suggesting that *V. goudotiana* Triana & Planch and *V. sphaerocarpa* (García-Barr. & Hern. Cam.) V.M. Badillo as well as *V. pubescens* A. DC. and *V. sprucei* (V.M. Badillo) V.M. Badillo are conspecific (Fig 2). This multilocus phylogeny also distinguished the split of two evolutionary lineages in *Vasconcellea*. One of these lineages (BS/BI = 67/0.97) is composed of 18 species, including two unidentified species (*Vasconcellea* sp. 1 and *Vasconcellea* sp. 2). These unidentified species were determined to have sister relationship to *V. monoica* (Desf.) A. DC. and *V. pubescens*/*V. sprucei*, respectively. The

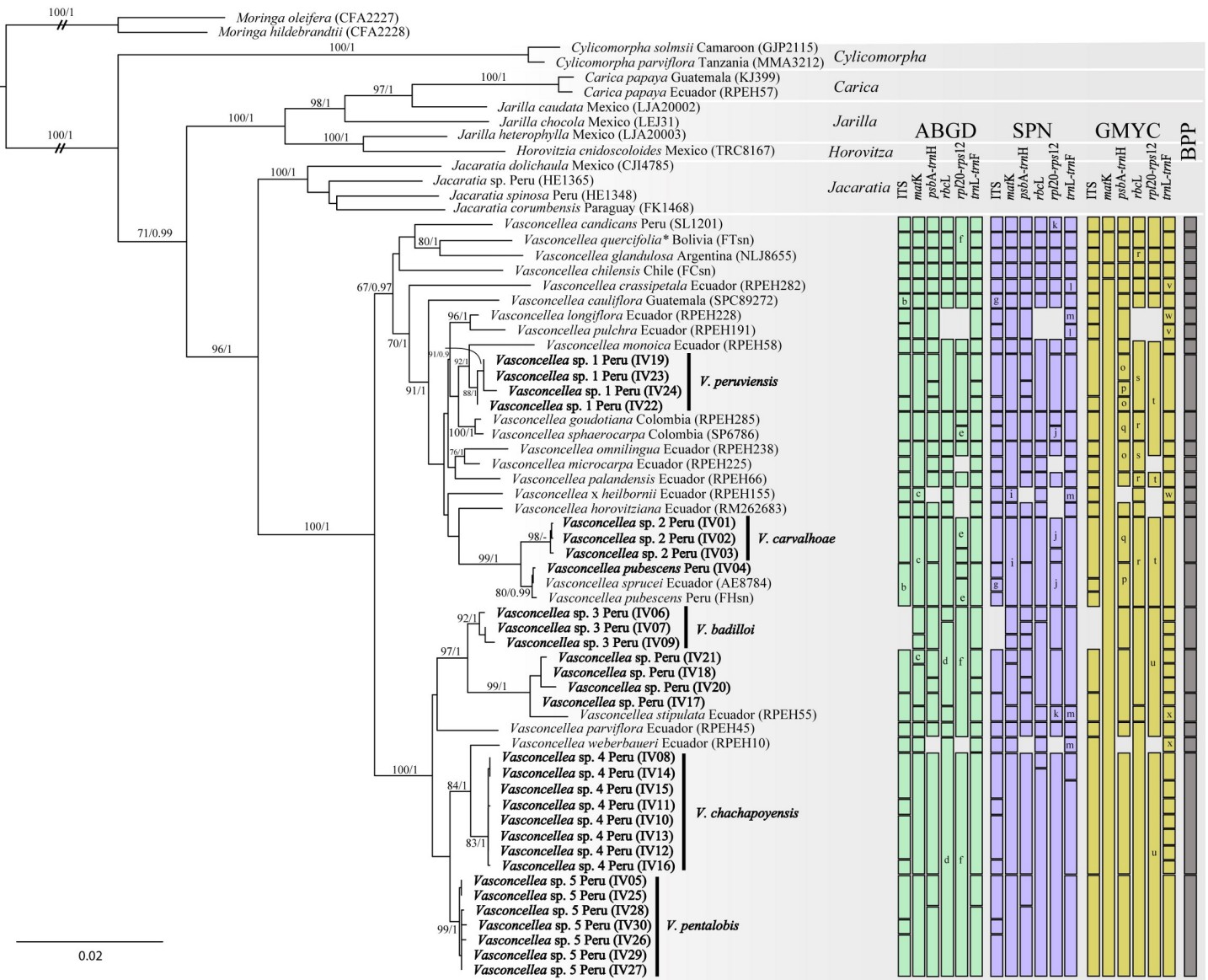

**Fig 2. Phylogenetic tree based on maximum likelihood inference of combined *matK, rbcL, trnL-trnF, psbA-trnH, rpl20-prs12,* and ITS data.** Value above branches = Maximum likelihood bootstrap values (BS)/Bayesian posterior probabilities (BI). Bars represent species delimitation results from ABGD-, SPN-, GMYC- and BPP-based algorithmic methods with six molecular markers. The scale bar indicates the number of nucleotide substitutions per site.

second lineage in *Vasconcellea* is a well-supported clade (BS/BI = 100/1.0) composed of six species, including three unidentified species (*Vasconcellea* sp. 3, *Vasconcellea* sp. 4, and *Vasconcellea* sp. 5). *Vasconcellea* sp. 3 was found to have sister relationship to *V. stipulata* (V.M. Badillo) V.M. Badillo, and these two taxa were sisters to *V. parviflora* A. DC. *Vasconcellea* sp. 4 was found to have sister relationship to *V. weberbaueri* (Harms) V.M. Badillo, and these two taxa were sisters to *Vasconcellea* sp. 5.

Additionally, the tree topologies for two to four loci (S1–S3 Figs) and individual marker showed incongruence (S4 Fig). These trees showed slight topological differences in the evolutionary relationships among genera of Caricaceae but strong distinctiveness among species in the genus *Vasconcellea*. For instance, different gene trees (e.g., *trnL-trnF*, S4 Fig) showed

**Table 3. Lowest genetic distance (p-distances) in percentage for species of *Vasconcellea* for six markers.**

| Taxa | Markers | | | | | |
|------|---------|---------|---------|------|---------|---------|
| | **ITS** | ***mat*K** | ***psb*A-*trn*H** | ***rbc*L** | ***rpl*20-*rps*12** | ***trn*L-*trn*F** |
| *V. cauliflora—V. sprucei* | 0.00 | 0.30 | 1.20 | 0.20 | - | 0.50 |
| *V. microcarpa—V. omnilingua* | 1.10 | 0.00 | 0.00 | 0.20 | - | 0.00 |
| *V. pubescens—V. omnilingua* | 0.40 | 0.20 | 0.00 | 0.30 | 0.50 | 0.50 |
| *V. glandulosa—V. pubescens* | 1.10 | 1.10 | 7.80 | 0.00 | 1.10 | 0.80 |
| *V. parviflora—V. stipulata* | 4.00 | 0.50 | 18.20 | 0.20 | 0.00 | 0.80 |
| *V. badilloi—V. stipulata* | - | 0.30 | 11.90 | 0.20 | 0.10 | 0.00 |
| *V. chachapoyensis—V. weberbaueri* | 1.10 | 0.60 | - | 0.00 | - | 0.30 |
| *V. pentalobis—V. chachapoyensis* | 1.10 | 0.30 | 6.00 | 0.00 | 0.00 | 0.50 |
| *V. peruviensis—V. monoica* | 0.40 | | 7.80 | 0.00 | 0.40 | 0.80 |
| *V. carvalhoae—V. pubescens* | 0.36 | 0.00 | - | 0.00 | 0.00 | 0.00 |

distinguishing interspecific relationships among several species of *Vasconcellea*, indicating that these species share different common ancestors depending on the marker, which is understood under hybridization scenarios [44]. In addition, the genetic divergence comparisons of the six markers showed that there is not a minimum threshold (p-distance) for distinguishing genetic species in *Vasconcellea* (Table 3) confirming genetic discordance. For instance, *V. microcarpa* (Jacq.) A. DC. and *V. omnilingua* (V.M. Badillo) V.M. Badillo are identical when comparing *matK*, *psbA-trnH*, and *trnL-trnF*; but different when comparing ITS, *rbcL*, and *rpl20-rps12* (Table 3).

## Species delimitation

The species-delimitation methods based on genetic distance (ABGD, SPN) and coalescence (GMYC, BPP) showed incongruent results for the six genes (Fig 2, Table 4). Among these methods, the highest number of species was delimited by the BPP analyses (25), whereas the most conservative results were obtained from GMYC (16 ± 10) (S5 and S6 Figs, S2 Table). Moreover, similar species numbers resulted from the ABGD (22 ± 6) and SPN (23 ± 5) analyses. The additional species delimitation by ABGD, BPP, and SPN was mainly due to the split of the clades composed of *V. pubescens*/*V. sprucei*, *V. stipulata*, *Vasconcellea* sp. 1, *Vasconcellea* sp. 2, *Vasconcellea* sp. 3, *Vasconcellea* sp. 4, and *Vasconcellea* sp. 5 (Fig 2). However, the split of these clades was not supported by the posterior probabilities obtained from BPP analyses (S3 Table). Although there were incongruent results in species number among different methods, the genetic distance methods (ABGD, SPN) and the multi-locus coalescent species validation (BPP) showed similar species numbers with those obtained in the phylogenetic analyses (Fig 2, Table 4). Regarding the six molecular makers, the highest number of species was delimited for the spacer ITS (27 ± 4) and the intergenic *trnL-trnF* (27 ± 6), whereas the lowest numbers were obtained for the genes *matK* (16 ± 10) and *rbcL* (17 ± 4) and the intergenic *rpl20-prs12* (12 ± 5) (Table 4). These low species numbers were a consequence of the merging of several species that have similar sequences with null or very low genetic distance between these markers as a consequence of hybridization events (Table 3).

## Taxonomic treatment

Our molecular analyses revealed that the five unidentified species of *Vasconcellea* were strongly supported as distinct entities by BPP and multilocus phylogeny and confirmed by ABGD (ITS, *psbA-trnH*, *trnL-trnF*) and SPN (ITS, *psbA-trnH*). This molecular examination of

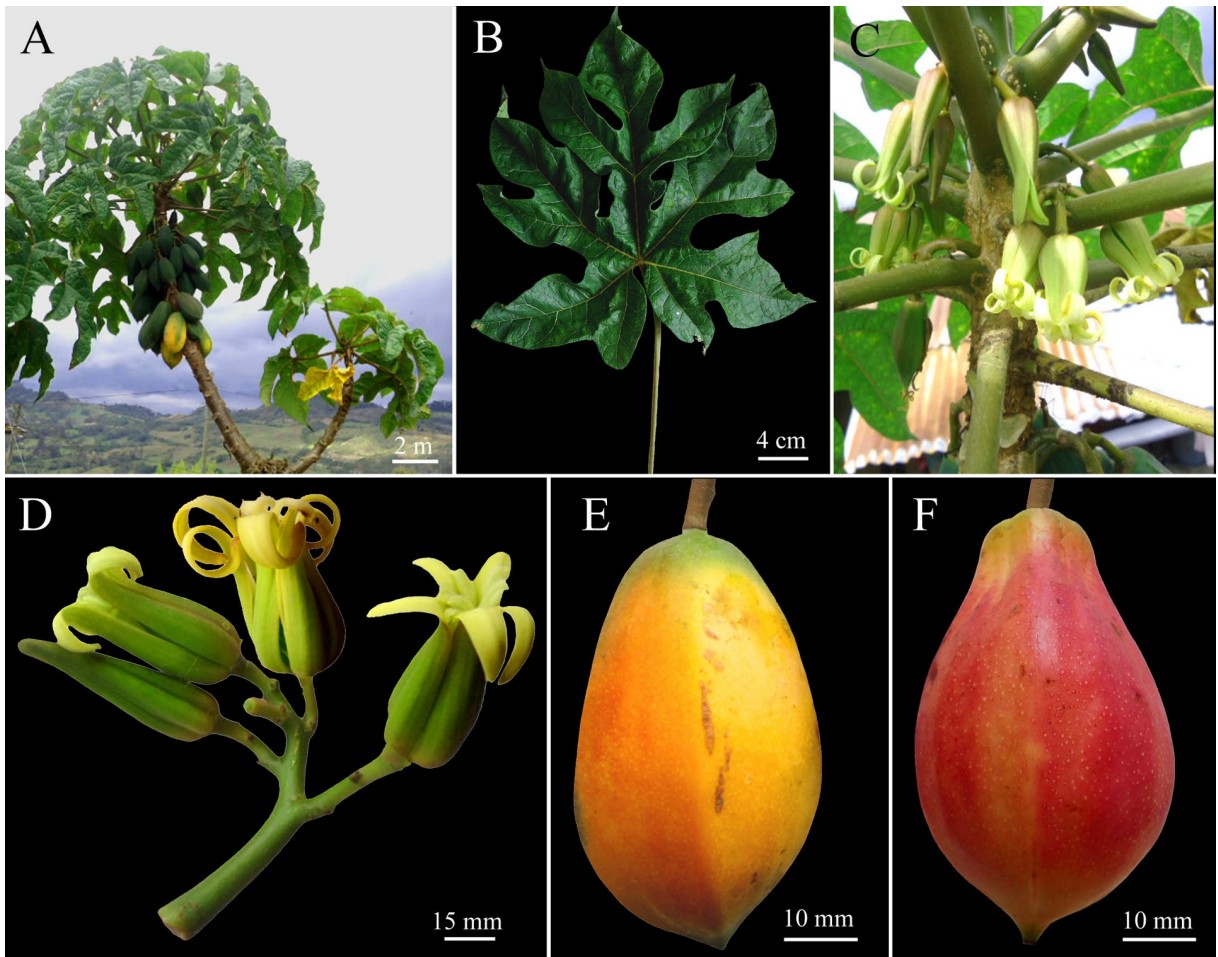

**Fig 3. Morphology of *Vasconcellea badilloi* sp. nov. (CHAX224). A,** Habit. **B,** Palmately compound leaf. **C,** Axillary flowers. **D,** Female inflorescence. **E, F,** Mature fruit.

*Vasconcellea* taxa based on an integrative approach justifies the proposal of the following five taxa as new species: *V. badilloi* sp. nov. (= *Vasconcellea* sp. 3), *V. carvalhoae* sp. nov. (= *Vasconcellea* sp. 2), *V. chachapoyensis* sp. nov. (= *Vasconcellea* sp. 4), *V. pentalobis* sp. nov. (= *Vasconcellea* sp. 5), and *V. peruviensis* sp. nov. (= *Vasconcellea* sp. 1)

***Vasconcellea badilloi*** D. Tineo & D.E. Bustam., **sp. nov.** (Fig 3)

[urn:lsid:ipni.org:names: 77212844–1]

**Holotype.**   Peru, Amazonas, Prov. Bongará, Dist. Pomacochas, 5˚48'53''S, 77˚57'24"W, 2280 m a.s.l., 13 Sep. 2018, *D. Tineo IV06* (holotype, CHAX224).

**Diagnosis.**   Dioecious tree to 6 m tall having as a distinguishing feature its yellow-orange to pink ovoid berries. Species very similar morphologically to *V. carvalhoae* and *V. pubescens* but differing in the phylogenetic relationship with these species. Furthermore, *V. badilloi* is distinguished genetically from *V. pubescens* (1.2% for *matK*, 8.4% for *psbA-trnH*, 0.5% for *rbcL*, 1.3% for *rpl20-rps12*, 0.4% for *trnL-trnF*) and *V. carvalhoae* (1.2% for *matK*, 8.4% for *psbA-trnH*, 0.5 for *rbcL*, 1.3% for *rpl20-rps12*, 0.3% for *trnL-trnF*).

**Description.**   Dioecious tree to 6 m tall (Fig 3A); bark light brown, covered with leaf scars; stipules absent. Latex white milky. Leaves roughly textured, alternate, crowded at top of tree, palmately compound (Fig 3B); petiole to 75 cm long; leaflets 5 to 7, glabrous and bright green

**Table 4. Species number in *Vasconcellea* identified with DNA-based species-delimitations methods and phylogeny.**

| Taxa | Genetic Distance | | | | | | | | | | | | Coalescence | | | | | | Genealogical Concordance |
|---|---|---|---|---|---|---|---|---|---|---|---|---|---|---|---|---|---|---|---|
| | ABGD | | | | | | SNP | | | | | | GMYC | | | | | | BPP |
| | ITS | matK | psbA-trnH | rbcL | rpl20-rps12 | trnL-trnF | ITS | matK | psbA-trnH | rbcL | rpl20-rps12 | trnL-trnF | ITS | matK | psbA-trnH | rbcL | rpl20-rps12 | trnL-trnF | Multilocus |
| ***V. badilloi*** | - | 2 | 1 | x | x | 1 | - | 2 | 3 | x | 1 | x | - | x | 1 | x | x | 3 | 1 |
| *V. candicans* | 1 | 1 | 1 | 1 | x | 1 | 1 | 1 | 1 | 1 | x | 1 | 1 | 1 | 1 | 1 | x | 1 | 1 |
| *V. cauliflora* | x | 1 | 1 | 1 | 1 | 1 | x | 1 | 1 | 1 | 1 | 1 | 1 | x | 1 | 1 | 1 | 1 | 1 |
| ***V. carvalhoae*** | 1 | x | 1 | x | x | 1 | 1 | x | 1 | x | x | x | x | x | x | x | x | x | 1 |
| ***V. chachapoyensis*** | 4 | 1 | 1 | x | x | 1 | 4 | 1 | 1 | x | x | 2 | x | x | 1 | x | x | 7 | 1 |
| *V. chilensis* | 1 | 1 | 1 | 1 | 1 | 1 | 1 | 1 | 1 | 1 | 1 | 1 | 1 | 1 | 1 | 1 | 1 | 1 | 1 |
| *V. crassipetala* | 1 | 1 | 1 | 1 | 1 | 1 | 1 | 1 | 1 | 1 | 1 | x | 1 | x | 1 | 1 | 1 | x | 1 |
| *V. pubescens / V. sprucei* | x | x | 1 | x | x | 1 | x | x | 1 | x | x | x | x | x | x | x | x | x | 1 |
| *V. glandulosa* | 1 | 1 | 1 | 1 | x | 1 | 1 | 1 | 1 | 1 | 1 | 1 | 1 | x | 1 | x | 1 | 1 | 1 |
| *V. goudotiana / V. sphaerocarpa* | 1 | 1 | 1 | 1 | x | 1 | 2 | 1 | 1 | 1 | x | 1 | 1 | x | x | x | x | 1 | 1 |
| *V. x heilbornii* | x | x | - | 1 | - | 1 | x | x | - | 1 | - | x | x | x | - | 1 | - | x | 1 |
| *V. horovitziana* | 1 | 1 | 1 | 1 | - | 1 | 1 | 1 | 1 | 1 | - | 1 | 1 | x | 1 | 1 | - | 1 | 1 |
| *V. longiflora* | 1 | x | x | - | - | x | 1 | x | x | - | - | x | 1 | x | x | - | - | x | 1 |
| *V. microcarpa* | 1 | x | 1 | 1 | - | 1 | 1 | x | 1 | 1 | - | 1 | 1 | x | x | x | - | 1 | 1 |
| *V. monoica* | 1 | 1 | 1 | x | 1 | 1 | 1 | 1 | 1 | x | 1 | 1 | x | x | 1 | x | x | 1 | 1 |
| *V. omnilingua* | 1 | x | 1 | 1 | 1 | 1 | 1 | x | 1 | 1 | 1 | 1 | 1 | x | x | x | x | 1 | 1 |
| *V. palandensis* | 1 | x | 1 | 1 | 1 | 1 | 1 | x | 1 | 1 | 1 | 1 | 1 | x | 1 | 1 | x | 1 | 1 |
| *V. parviflora* | 1 | 1 | 1 | 1 | x | 1 | 1 | 1 | 1 | 1 | 1 | 1 | 1 | x | 1 | x | x | 1 | 1 |
| ***V. pentalobis*** | 3 | 1 | 2 | x | x | 2 | 3 | 1 | 2 | x | x | 1 | 1 | x | 1 | x | x | 1 | 1 |
| ***V. peruviensis*** | 2 | 1 | 3 | x | 1 | 3 | 2 | 1 | 3 | x | 1 | 1 | x | x | x | x | x | 1 | 1 |
| *V. pulchra* | 1 | x | x | - | - | x | 1 | x | x | - | - | x | 1 | x | x | - | - | x | 1 |
| *V. quercifolia* | 1 | 1 | 1 | 1 | x | 1 | 1 | 1 | 1 | 1 | 1 | 1 | 1 | x | 1 | 1 | x | 1 | 1 |
| *V. stipulata* | 1 | x | 1 | x | x | 1 | 1 | x | 1 | x | x | x | 1 | x | x | x | x | x | 1 |
| *V. weberbaueri* | 1 | 1 | - | x | - | 1 | 1 | 1 | - | 1 | - | x | x | x | - | x | - | x | 1 |
| ***Vasconcellea* sp.** | x | x | 2 | x | x | 2 | x | x | 2 | x | x | x | x | x | x | x | x | 3 | 1 |
| Total | 28 | 21 | 26 | 18 | 12 | 28 | 31 | 22 | 28 | 21 | 17 | 20 | 23 | 4 | 18 | 12 | 7 | 32 | 28 |

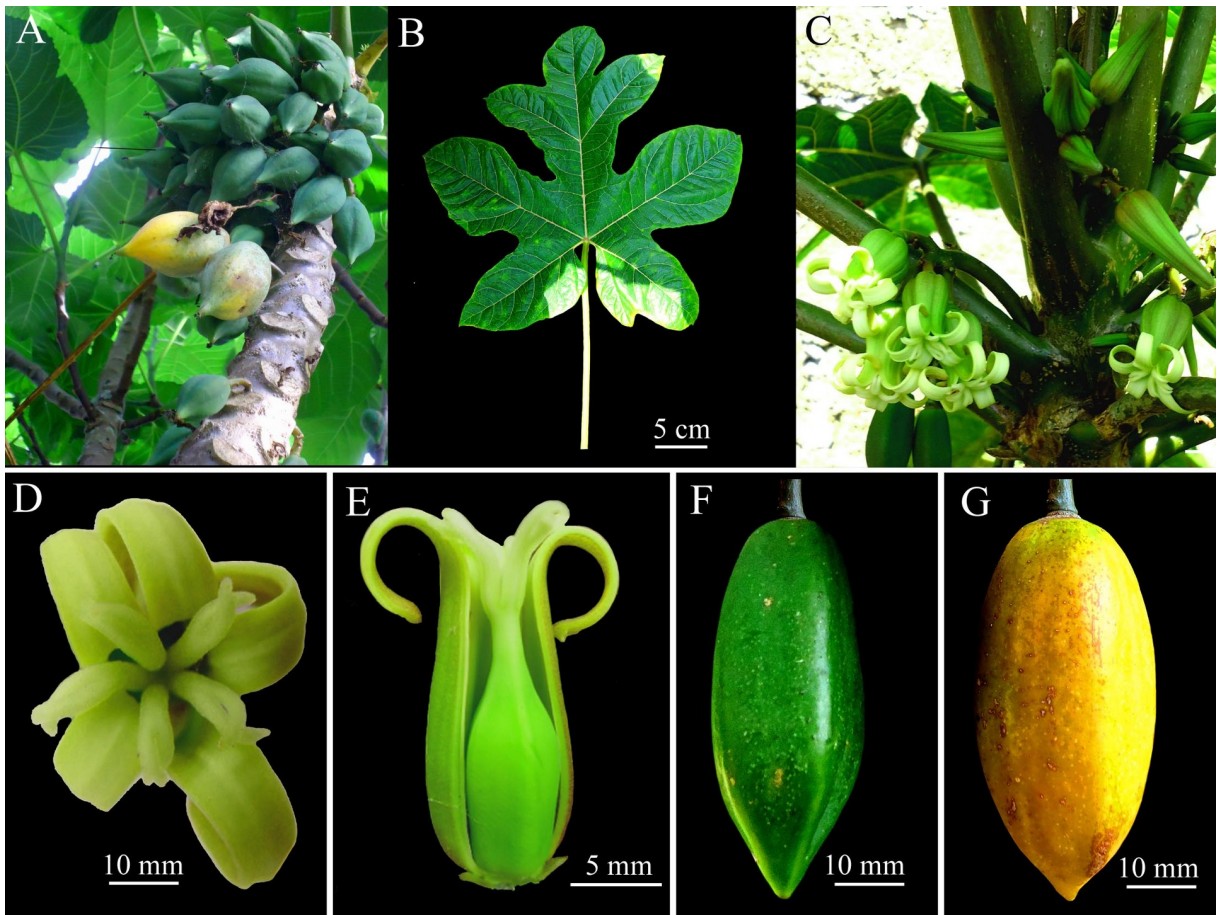

**Fig 4. Morphology of *Vasconcellea carvalhoae* (CHAX227). A,** Habit. **B,** Palmately compound leaf. **C,.** Female inflorescence. **D, E,** Female flowers with stigmas split into 2–3 ends (D) and superior ovaries (E). **F,** Immature fruit. **G,** Mature fruit.

above, lighter green below; 4 to 5 basal leaflets entire 18–30 × 7–9 cm, widely elliptic to widely ovate, base acute, apex acuminate; central leaflet trilobed, 2 lateral lobes 12–15 × 3.5–5 cm, elliptic to ovate, apex acuminate, central lobe 20–35 × 4–8 cm, elliptic to ovate, base acute, apex acuminate; veins raised beneath, primary vein often reddish. Female inflorescences axillary (Fig 3C and 3D), cymose, 4 to 5 cm long; peduncle 1–2 cm long, 3 mm diam.; pedicels 2–3 mm long, with a few small bracts 1 mm long. Female flowers 5-merous. Sepals green, triangular, 2–3 × 1–2 mm. Petals green-yellowish outside, green inside, free, oblong-obtuse, 25–35 × 5–7 mm, apex obtuse. Sepals and petals alternate. Ovary superior, 5-locular, 5-angular, 9–18 × 5–8 mm, attenuate towards apex; numerous anatropous ovules on parietal placentas; style 4–5 mm long; stigmas 5, 5–7 mm long, apically often split in 2 ends of 2–3.5 mm each. Male inflorescences were not found in this study. Fruit an ovoid berry, yellow-orange to pink, base rounded to emarginate, apex acute, 60–75 × 4–5.5 mm; pericarp 4–5 mm thick; pedicel of fruit 9–12 × 4–5 mm (Fig 3E and 3F). Seeds dark brown, 7–8 × 4–5 mm, ellipsoidal, sclerotesta with conical protuberances, each seed surrounded by a gelatinous sarcotesta, seeds arranged in 5 groups surrounded by yellowish pulp. Strong aroma. The sugar content varies from 6 to 7°Brix.

**Etymology.** The specific epithet '*badilloi*' honours Victor M. Badillo for his pioneering and valuable contributions to the understanding of Caricaceae in South America, especially in the genus *Vasconcellea*.

**Ecology and distribution.** The species is known from the area around Pomacochas (5˚ 48'53"S 77˚57'24"W) in the province of Bongará (Amazonas, Peru) and around Quinjalca (6˚ 05'33"S 77˚40'39"W) in the province of Chachapoyas (Amazonas, Peru). It is found in the wild in wet premontane to montane forests at 1300–3200 m elevation. Plants are cultivated.

**Specimens examined.** Peru, Amazonas, Prov. Bongará, Dist. Cuchulia, 5˚59'44"S, 77˚ 58'30'W', 1386 m a.s.l., 13 Sep. 2018, *D. Tineo IV07* (CHAX225); Peru, Amazonas, Prov. Chachapoyas, Dist. Quinjalca, 6˚05'33"S, 77˚40'39"W, 3143 m a.s.l., 20 Sep. 2018, *D. Tineo IV09* (CHAX226).

**Remarks.** *Vasconcellea badilloi* highly resembles *V. carvalhoae* and *V. pubescens* in morphology, growing in sympatry. However, *V. badilloi* is distinguished by its yellow-orange to pink ovoid berries. *V. badilloi* is also distinguished from *V. pubescens* by the lack of pubescence on the leaves. Furthermore, *V. badilloi* is distantly related to *V. carvalhoae* and *V. pubescens* with multilocus phylogeny. In addition, although *V. badilloi* is phylogenetically closely related to *V. stipulata*, *V. badilloi* is distinguished from *V. stipulata* by the lack of stipules (S4 Table).

*Vasconcellea carvalhoae* D. Tineo & D.E. Bustam., **sp. nov**. (Fig 4)

[urn:lsid:ipni.org:names: 77212845–1]

**Holotype.** Peru, Amazonas, Prov. Bongará, Dist. Pomacochas, 5˚49'45"S, 77˚58'12"W, 2232 m a.s.l., 02 Oct. 2018, *D. Tineo IV01* (holotype, CHAX227).

**Diagnosis.** Dioecious tree to 4 m tall that is very similar morphologically to *V. sprucei*/*V. pubescens*, but differing in the sister phylogenetic relationship with these species. The sequence divergence between *V. carvalhoae* and *V. sprucei*/*V. pubescens* is 0.36% for the ITS region.

**Description.** Dioecious tree to 4 m tall; bark light brown, covered with leaf scars. Latex white milky (Fig 4A). Leaves membranaceus, alternate, crowded at top of tree, palmately compound; petiole to 60 cm long (Fig 4B); leaflets 5, glabrous and bright green above, lighter green below; 4 basal leaflets entire, 20–30 × 10–15 cm, widely elliptic to widely ovate, base acute, apex acute to rounded; central leaflet trilobed, 2 lateral lobes 17–19 × 7–10 cm, widely elliptic to widely ovate, apex acute to rounded, central lobe 35–40 × 10–12 cm, widely elliptic to widely ovate, apex acute to rounded; veins raised beneath, primary vein often reddish. Female inflorescences axillary (Fig 4C), cymose, many-flowered, to 16 cm long; peduncle 5–7 cm long, 4–7 mm diam.; pedicels 2–3 cm long, with a few small bracts 1 mm long. Female flowers 5-merous. Sepals green, triangular, 2–3.5 × 1.5–2 mm. Petals green-yellow outside, green inside, free, oblong-obtuse, 30–40 × 5.5–7 mm, apex obtuse. Sepals and petals alternate. Ovary superior, 5-locular, 5-angular, 10–20 × 6–9 mm, attenuate towards apex; numerous anatropous ovules on parietal placentas; style 3–4 mm long; stigmas 5, 5–8 mm long, apically often split in 2 ends of 3–5 mm each (Fig 4D and 4E). Fruit an ovoid berry (Fig 4F and 4G), yellow, base rounded to emarginate, apex acute, 60–80 × 42–56 mm; pericarp 4–5.5 mm thick; pedicel of fruit 10– 17 × 4.5–5 mm. Seeds light brown, 5–7 × 3–5 mm, ellipsoidal, sclerotesta with numerous conical protuberances, each seed surrounded by a gelatinous sarcotesta, seeds arranged in 5 groups surrounded by yellow-white pulp. The sugar content varies from 7.5 to 8˚Brix.

**Etymology.** The specific epithet '*carvalhoae*' honours Fernanda A. Carvalho for her valuable contributions to the understanding of the Caricaceae in the bioinformatics era.

**Ecology and distribution.** The species is known from the area around Pomacochas (5˚ 49'45"S 77˚58'12"W) in the province of Bongará, Amazonas, Peru. It is found in the wild in montane areas at 2236 m elevation. Plants not cultivated.

**Specimens examined.** Peru, Amazonas, Prov. Bongará, Dist. Pomacochas, 5˚49'08.7"S, 77˚57'39.3"W, 2263 m a.s.l., 02 Oct. 2018, *D. Tineo IV02* (CHAX228); Peru, Amazonas, Prov. Bongará, Dist. Pomacochas, 5˚49'37"S, 77˚58'01"W, 2236 m a.s.l., 02 Oct. 2018, *D. Tineo IV03* (CHAX229).

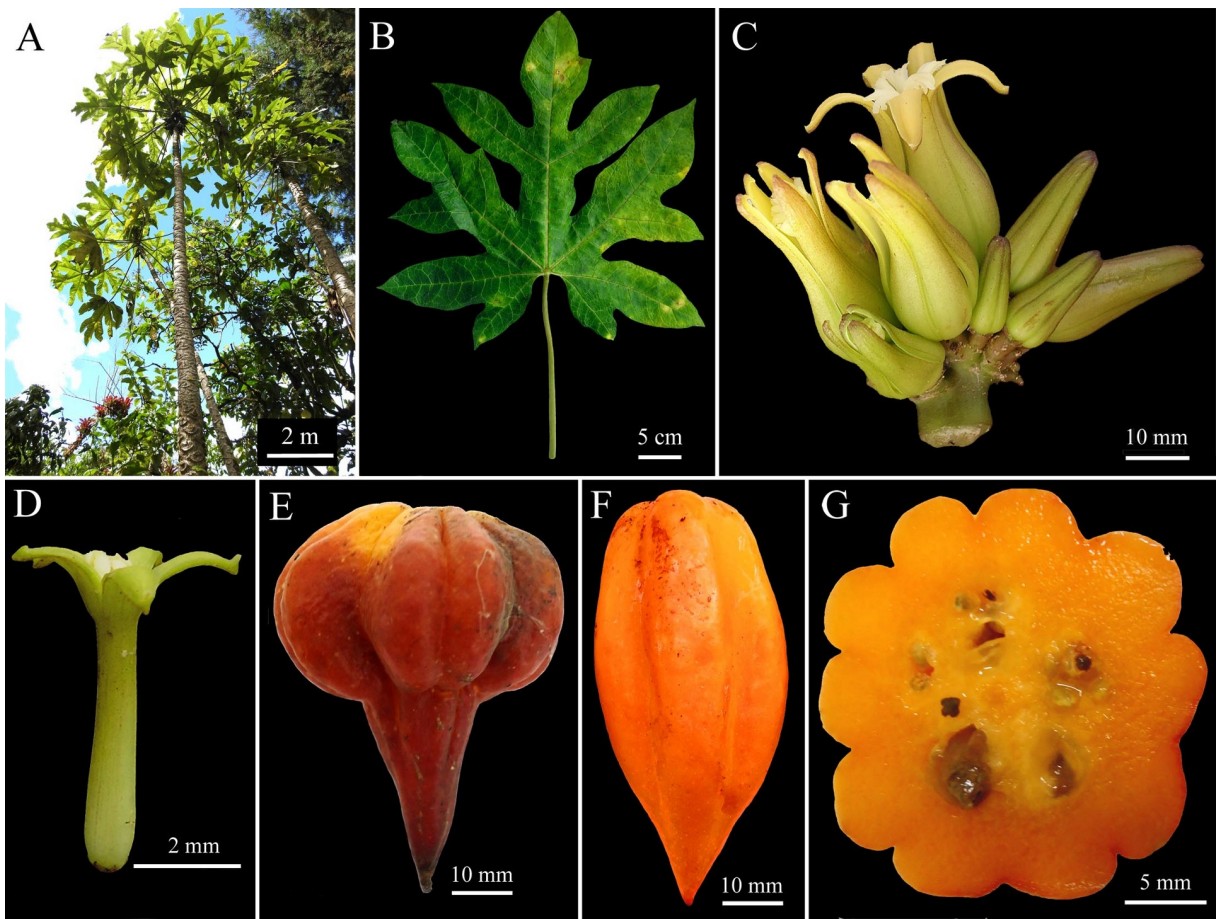

**Fig 5. Morphology of *Vasconcellea chachapoyensis* sp. nov. (CHAX235). A,** Habit. **B,** Palmately compound leaf. **C,** Female inflorescence. **D,** Male flower. **E, F,** Mature fruits with acuminate apex. **G,** 5-locular ovary.

**Remarks.** *Vasconcellea carvalhoae* is highly similar in morphology to *V. sprucei*/*V. pubescens*, growing in sympatry. However, these species are distinguished by their elongated ovoid berry and lack of pubescence (S4 Table). Phylogenetically, *V. carvalhoae* is also a sister to the clade composed of *V. sprucei*/*V. pubescens*. They are genetically different species based on a 0.36% divergence in the ITS region.

***Vasconcellea chachapoyensis*** D. Tineo & D.E. Bustam., **sp. nov.** (Fig 5)

[urn:lsid:ipni.org:names: 77212846–1]

**Holotype.** Peru, Amazonas, Prov. Chachapoyas, Dist. Asunción Goncha, 6˚01'56.6"S, 77˚42'37.1"W, 2821 m a.s.l., 20 Sep. 2018, *D. Tineo IV14* (holotype, CHAX235).

**Diagnosis.** Dioecious tree to 12 m tall having ovoid berries with acuminate apex as distinguishing features. Species very similar morphologically to *V. weberbaueri*, but differing in the larger inflorescence and the wider leaflets at the base and in the sister phylogenetic relationship with this species. The sequence divergence between *V. chachapoyensis* and *V. weberbaueri* is 1.1% for the ITS, 0.6% for *matK*, and 0.3% for *trn*L-*trn*F.

**Description.** Dioecious tree to 12 m tall (Fig 5A); bark light brown, covered with prominent leaf scars; stipules absent. Latex white milky. Leaves membranaceus, alternate, crowded at top of tree, palmately compound, deeply lobe (Fig 5B); petiole 50 to 70 cm long; leaflets 5 to 7, glabrous and bright green above, lighter green below with purple red stripes and stipules in the

veins; 4 to 7 basal leaflets entire, 9.5–30.2 × 6.6–12.4 cm, widely elliptic to ovate, base obtuse, apex acuminate; central leaflet trilobed, 2 lateral lobes 12–18.5 × 4.5–7.6 cm, widely elliptic to ovate, apex acuminate, central lobe 25–38 × 8–10.7 cm, elliptic to ovate, base acute, apex acuminate (Fig 5B). Female inflorescences axillary (Fig 5C), cymose, few-flowered, to 9 cm long; peduncle 3–4 cm long, 4–5 mm diam.; pedicels 5–15 mm long, with a few small bracts 1.5 mm long. Female flowers 5-merous. Sepals greenish, triangular, 3–4 × 2–3 mm. Petals green-yellowish, green inside, free, oblong-triangular, 31–42 × 6–9 mm, apex obtuse. Sepals and petals alternate. Ovary superior, 5-locular, 5-angular, 6–8 × 3–4 mm, attenuate towards apex; numerous anatropous ovules on parietal placentas; style 5–8 mm long; stigmas 5, 7–12 mm long, short, apically often split in 2–4 ends of 2–4 mm each. Male inflorescences axillary, many-flowered panicles, to 25–35 cm long, pubescent; peduncle 15–20 cm long, to 4 mm diam.; lateral branches 3–6 cm long; pedicels 2–6 mm long, with a few small bracts to 1 mm long. Male flowers 5-merous (Fig 5D). Sepals brown, triangular, 2–3 × 1–2 mm. Corolla green-yellow; tube 16–23 mm long, 3–4 mm wide at base, 1.5–2 mm wide in the middle, 2.5–4 mm wide at apex; lobes oblong-lanceolate, 14–20 × 2–3 mm, apex acute. Sepals and petals alternate. Stamens 10, in 2 series, attached at apex of corolla tube, versatile, 2 thecae each, opening with longitudinal slits, introrse; upper stamens with loosely pilose filaments 2–2.4 mm long, anther glabrous, 2–2.5 mm long, anther connective; lower stamens with filament 1–1.5 mm long, anther glabrous, 2–3 mm long. Rudimentary gynoecium 9 to 11 mm long. Fruit an ovoid berry, yellow-orange, base emarginate, apex acuminate, 75–85 × 35–45 mm; pericarp 12–15 mm thick; pedicel of fruit 3–4 × 1–3 mm (Fig 5E and 5F). Seeds light brown, 4–5 × 3–4 mm, ellipsoidal, sclerotesta with numerous conical protuberances, each seed surrounded by a gelatinous sarcotesta, seeds arranged in 5 groups surrounded by orange pulp. Starry central cavity (Fig 5G). The sugar content varies from 6 to 6.5˚Brix.

**Etymology.** The specific epithet '*chachapoyensis*' is derived from the province where the samples were collected.

**Ecology and distribution.** The species is known from the area around Chachapoyas (6˚ 01'56.6"S 77˚42'37.1"W) in the Region Amazonas, Peru. It is found in the wild in humid montane forest at 2400–3800 m elevation. Plants are not cultivated.

**Specimens examined.** Peru, Amazonas, Prov. Chachapoyas, Dist. Quinjalca, 6˚05'30.4"S, 77˚40'30.4"W, 3130 m a.s.l., 20 Sep. 2018, *D. Tineo IV08* (CHAX230); Peru, Amazonas, Prov. Chachapoyas, Dist. Quinjalca 6˚05'25"S, 77˚40'46"W, 3150 m a.s.l., 20 Sep. 2018, *D. Tineo IV15* (CHAX236); Peru, Amazonas, Prov. Chachapoyas, Dist. Olleros, 6˚03'07"S, 77˚38'54"W, 3041 m a.s.l., 20 Sep. 2018, *D. Tineo IV11* (CHAX232); Peru, Amazonas, Prov. Chachapoyas, Dist. Olleros, 6˚03'13.2"S, 77˚38'47.3"W, 3031 m a.s.l., 20 Sep. 2018, *D. Tineo IV12* (CHAX233); Peru, Amazonas, Prov. Chachapoyas, Dist. Granada, 6˚06'12"S, 77˚37'47"W, 2996 m a.s.l., 20 Sep. 2018, *D. Tineo IV10* (CHAX231); Peru, Amazonas, Prov. Chachapoyas, Dist. Granada, 6˚06'10"S, 77˚37'39"W, 3017 m a.s.l., 20 Sep. 2018, *D. Tineo IV13* (CHAX234); Peru, Amazonas, Prov. Chachapoyas, Dist. Molinopampa, 6˚16'59.2"S, 77˚33'31.7"W, 2200 m a.s.l., 25 Sep. 2018, *D. Tineo IV16* (CHAX237).

**Remarks.** *Vasconcellea chachapoyensis* is highly similar in morphology to *V. weberbaueri*, but it is distinguished by its larger inflorescence and wider leaflets at the base (S4 Table). Phylogenetically, *V. chachapoyensis* is also a closely related species to *V. weberbaueri*. However, these two species are genetically different at the ITS (1.1%), *matK* (0.6%), and *trnL-trn*F (0.3%) loci. Additionally, *V. chachapoyensis* grows in sympatry with *V. pentalobis*.

***Vasconcellea pentalobis*** D. Tineo & D.E. Bustam., **sp. nov.** (Fig 6)
[urn:lsid:ipni.org:names: 77212847–1]

**Holotype.** Peru, Amazonas, Prov. Chachapoyas, Dist. Molinopamapa, 6˚14'49"S 77˚ 32'50"W, 2297 m a.s.l., 05 May. 2018, *D. Tineo IV05* (holotype, CHAX238).

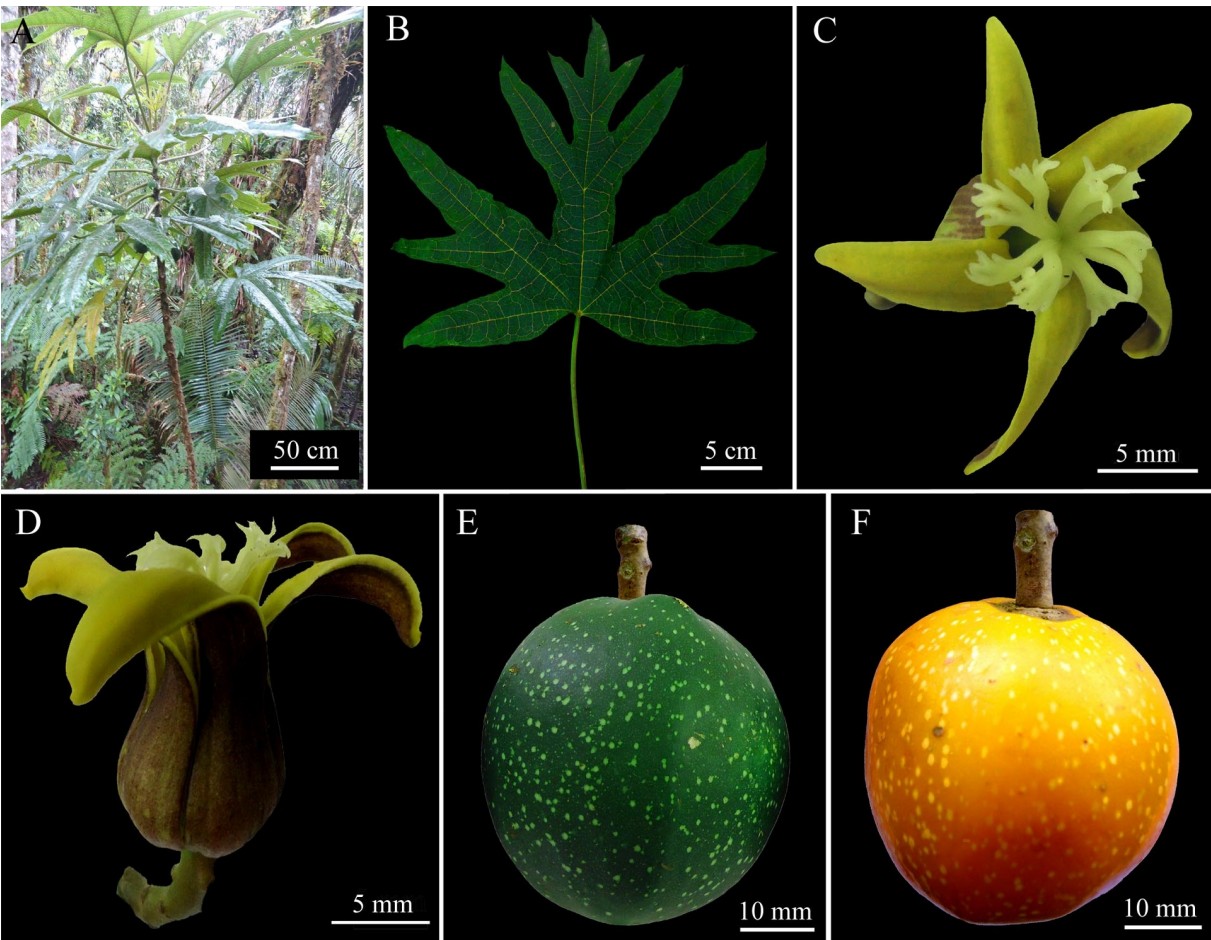

**Fig 6. Morphology of *Vasconcellea pentalobis* sp. nov. (CHAX238). A,** Habit. **B,**. Palmately compound leaf with four basal leaflets and a central pentalobed leafleft. **C,** Female flower with stigmas split in 2–3 ends. **D,** Dextrorotatory female flower. **E,** Immature fruit. **F,** Mature fruit.

**Diagnosis.** Dioecious tree to 8 m tall that is morphologically distinguished from any other species of *Vasconcellea* by the pentalobed central leaflet (four laterals and one central), dextrorotatory female flowers, and globose berries.

**Description.** Dioecious tree to 8 m tall (Fig 6A); bark light brown, covered with leaf scars; stipules absent. Latex white milky. Leaves membranaceus, alternate, slightly crowded at top of tree, palmately compound (Fig 6B); petiole to 40 to 50 cm long; leaflets 5 to 6, glabrous and bright green above, lighter green below with primary veins often green; usually 4 basal leaflets entire 30–39 × 10–13.5 cm, elliptic, base acute, apex acute; central leaflet pentalobed, 4 lateral lobes 15–35 × 4–8 cm, elliptic, apex acute, central lobe 10–18 × 3.5–6 cm, elliptic to ovate, base acute, apex acuminate (Fig 6B). Female inflorescences axillary, cymose, few-flowered, to 9 cm long, dextrorotatory (Fig 6C and 6D); peduncle 3.5–5 cm long, 3–4.5 mm diam.; pedicels 4.5–7.5 mm long, with a few small bracts 0.5–1.7 mm long. Female flowers 5-merous. Sepals green-reddish, triangular, 2–3 × 1–2 mm. Petals yellowish, free, oblong-triangular, 40–50 × 7–9 mm, apex obtuse. Sepals and petals alternate. Ovary superior, 5-locular, 5-angular, 9–15 × 6–12 mm, attenuate towards apex; numerous anatropous ovules on parietal placentas style 4–6 mm long stigmas 5, 5–7 mm long, apically often split in 2–3 ends of 0.5–2 mm each (Fig 6C). Male inflorescences were not found in this study. Fruit a globose berry, yellow-greenish with white

dots, base slightly flattened, apex rounded, 50–65 × 50–60 mm; pericarp 11–15 mm thick; pedicel of fruit 6–8 × 3–5 mm (Fig 6E and 6F). Seeds light brown, 4–5 × 3.5–4 mm, ellipsoidal, sclerotesta with small and numerous conical protuberances, each seed surrounded by a gelatinous sarcotesta, arranged in 5 groups surrounded by intense orange pulp. Semistarred central cavity. The sugar content varies from 7.5 to 8˚Brix.

**Etymology.** The specific epithet '*pentalobis*' refers to the diagnostic feature of the central leaflets that is composed of five lobes (four laterals and one central lobe).

**Ecology and distribution.** The species is known from the area around Chachapoyas (6˚ 14'49''S 77˚32'50''W) in the Region Amazonas, Peru. It is found in the wild in humid montane forest at 1600–2800 m elevation. Plants are not cultivated.

**Specimens examined.** Peru, Amazonas, Prov. Chachapoyas, Dist. Molinopampa, 6˚ 15'35''S, 77˚32'45''W, 2406 m a.s.l., 5 May. 2018, *D. Tineo IV25* (CHAX239); Peru, Amazonas, Prov. Chachapoyas, Dist. Molinopampa, 6˚20'08''S, 77˚31'49''W, 2385 m a.s.l., 20 Nov. 2019, *D. Tineo IV28* (CHAX242); Peru, Amazonas, Prov. Chachapoyas, Dist. Molinopampa, 6˚ 20'41''S, 77˚31'17''W, 2321 m.a.s.l., 20 Nov. 2019, *D. Tineo IV29* (CHAX243); Peru, Amazonas, Prov. Chachapoyas, Dist. Molinopampa, 6˚18'59''S, 77˚33'21''W, 2659 m a.s.l., 20 Nov. 2019, *D. Tineo IV30* (CHAX244); Peru, Amazonas, Prov. Chachapoyas, Dist. La jalca, 6˚28'25''S, 77˚ 42'13''W, 2523 m a.s.l., 15 Nov. 2019, *E. Huaman IV26* (CHAX240); Peru, Amazonas, Prov. Chachapoyas, Dist. La Jalca, 6˚28'25''S, 77˚41'51''W, 2342 m a.s.l., 15 Nov. 2019, *E. Huaman IV27* (CHAX241).

**Remarks.** *Vasconcellea pentalobis* is morphologically distinguished from any other species of *Vasconcellea* by the pentalobed central leaflet (four laterals and one central), dextrorotatory female flowers, and globose berries (S4 Table). *V. pentalobis* is sister to the clade composed of *V. chachapoyensis* and *V. weberbaueri* with our multilocus phylogeny. Additionally, *V. pentalobis* grows in sympatry with *V. chachapoyensis*.The genetic divergence between *V. pentalobis* and *V. chachapoyensis* is 1.1% for ITS, 0.3% for *matK*, 3.0% for *psb*A-*trn*H, and 0.6% for *trn*L-*trn*F, whereas that between *V. pentalobis* and *V. weberbaueri* is 1.4% for the ITS, 0.3% for *matK* and 0.3% for *trn*L-*trn*F

***Vasconcellea peruviensis*** D. Tineo & D.E. Bustam., **sp. nov.** (Fig 7)

[urn:lsid:ipni.org:names: 77212848–1]

**Holotype.** Peru, Amazonas, Prov. Utcubamba, Dist. Cajaruro, 5˚40'04''S 78˚20'17''W, 1538 m.a.s.l., 19 Oct. 2018, *D. Tineo IV23* (holotype, CHAX247).

**Diagnosis.** Monoecious tree to 4 m tall that is very similar morphologically to *V. monoica*, but differing by its globose berries with acute apices and in the sister phylogenetic relationship with this species. The sequence divergence between *V. peruviensis* and *V. monoica* is 0.4% for the ITS, 0.3–0.4% for *matK*, 4.6–5% for *psb*A-*trn*H, 0.3–0.4% for *rpl*20-*rps*12, and 0.6–0.8% for *trn*L-*trn*F.

**Description.** Monoecious tree to 4 m tall (Fig 7A); bark light brown and glabrous; slightly covered with leaf scars. Latex white milky. Leaves membranaceus, alternate, crowded at top of the stem, palmately compound (Fig 7B); petiole to 35 cm long; leaflets 5, glabrous and bright green above, lighter green below; 4 basal leaflets entire, 12–16 × 4–7 cm elliptic, base acute, apex acute to acuminate; central leaflet trilobed, 2 lateral lobes 9–12 × 3–5 cm, elliptic, apex acuminate, central lobe 20–25 × 5–7 cm, elliptic to ovate, base acute, apex acuminate; elevated veins below, primary vein green-reddish. Female and male inflorescences axillary grouped in few-flowered panicles, to 5 cm long; peduncle 1–3 cm long, to 2–3 mm diam.; lateral branches 1–3 cm long; pedicels 1–3 mm long, with a few small bracts to 1 mm long (Fig 7C). Female flowers 5-merous (Fig 7D). Sepals green, triangular, 1.5–2 × 0.5–1 mm. Petals yellow-white outside and inside, free, oblong-triangular, 33–45 × 5–7 mm, apex obtuse. Sepals and petals alternate. Ovary superior, 5-locular, 5-angular, 9–15 × 7–10 mm, attenuate towards apex;

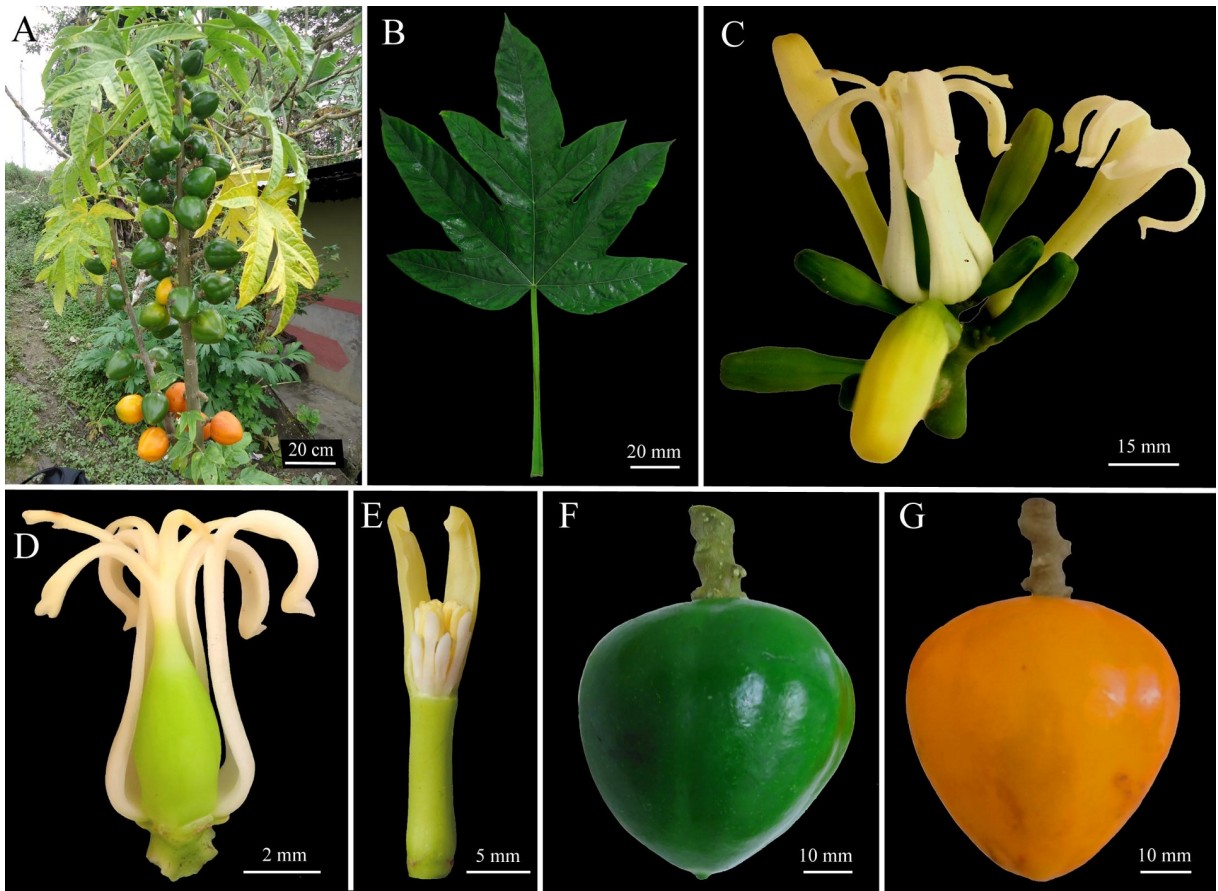

**Fig 7. Morphology of *Vasconcellea peruviensis* (CHAX245). A,** Habit. **B,** Palmately compound leaf. **C,** Female inflorescence. **D,** Pistillate flower. **E,** Staminate flower. **F,** Immature fruit. **G,** Mature fruit.

numerous anatropous ovules on parietal placentas; style 3–4.5 mm long; stigmas 5, 7–12 mm long, apically often split in 2 ends of 3–4 mm each. Male flowers 5-merous (Fig 7E). Sepals green, triangular, 1–2 × 0.5–1 mm. Corolla yellow-white; tube 18–22 mm long, 2–4 mm wide at base, 1.5–3 mm wide in the middle, 2–3.5 mm wide at apex; lobes oblong-lanceolate, 16–23 × 2–5 mm, apex acute. Sepals and petals alternate. Stamens 10, in 2 series, attached at the apex of corolla tube, versatile, 2 thecae each, opening with longitudinal slits, introrse; upper stamens with pilose filaments 1.5–2 mm long, anther glabrous, 2–3 mm long; lower stamens with filament 1 mm long, anther glabrous, 2–3 mm long, anther connective prolonged for 1 mm. Rudimentary gynoecium 6–9 mm long. Fruit a globose berry, yellow-orange, base rounded, apex acute, 10–18 to 6–7 cm; pericarp 4–5 mm thick; pedicel of fruit 6–9 × 4–5 mm (Fig 7F and 7G). Seeds dark brown, 6–8 × 4–5.5 mm, ellipsoidal, sclerotesta with numerous large and conical protuberances, each seed surrounded by a gelatinous sarcotesta, seeds arranged in 5 groups surrounded by yellow-white pulp. Superficial or low depression ridges. The sugar content varies from 4.5 to 5˚Brix.

**Etymology.** The specific epithet '*peruviensis*' is derived from the country where the samples were collected.

**Ecology and distribution.** The species is known from the area around the provinces of Utcubamba (5˚40'33.1"S 78˚20'23.8"W), Rodríguez de Mendoza (6˚26'35.4"S 77˚28'44.9"W),

and Chachapoyas (6˚31'33.0"S 77˚48'50.2"W). It is found in the wild in premontane wet forests and montane forests at 1200–1800 m elevation. Plants not cultivated.

**Specimens examined.** Peru, Amazonas, Prov. Utcubamba, Dist. Cajaruro, 5˚40'33.1"S, 78˚20'23.8"W, 1571 m a.s.l., 19 Oct. 2018, *D. Tineo IV19* (CHAX245); Peru, Amazonas, Prov. Rodríguez de Mendoza, Dist. Santa Rosa, 6˚26'35.4"S, 77˚28'44.9"W, 1887 m a.s.l., 15 Sep. 2018, *D. Tineo IV24* (CHAX248); Peru, Amazonas, Prov. Prov. Chachapoyas, Dist. La Jalca, 6˚31'33.0"S, 77˚48'50.2"W, 2557 m a.s.l., 01 Sep. 2018, *D. Tineo IV22* (CHAX246).

**Remarks.** Although *Vasconcellea peruviensis* is morphologically similar to *V. monoica*, this species is distinguished from other species of *Vasconcellea* by being monoecious trees. *V. peruviensis* is morphologically different from *V. monoica* by having globose berries with an acute apex (S4 Table). Additionally, *V. peruviensis* grows in sympatry with *V. pentalobis* and *V. stipulata*. Phylogenetically, *V. peruviensis* is also a closely related species to *V. monoica*. However, these two species are genetically different at the ITS (0.4%), *matK* (0.3–0.4%), *psbA-trnH* (4.6–5%), *rpl20-rps12* (0.3–0.4%), and *trnL-trnF* (0.6–0.8%) loci.

Morphologycally, the species of *Vaconcellea* reported from Peru can be distinguished by the following taxonomic key:

1a. Monoecious plants. . . . . . . . . . . . . . . . . . . . . . . . . . . . . . . . . . . . . . . . . . . . . . . . . . . . . . . . . . **2**

1b. Dioecious plants. . . . . . . . . . . . . . . . . . . . . . . . . . . . . . . . . . . . . . . . . . . . . . . . . . . . . . . . . **3**

2a. Petiole length 10–25 cm, seed surface having acute projections . . . . . . . . . . . . . . . .***V. monoica***

2b. Petiole length 25–35 cm, seed surface having large and conical protuberances
. . . . . . . . . . . . . . . . . . . . . . . . . . . . . . . . . . . . . . . . . . . . . . . . . . . . . . .***V. peruviensis***

3a. Deciduous plants . . . . . . . . . . . . . . . . . . . . . . . . . . . . . . . . . . . . . . . . . . . . . . . . . . . . . . . . ...**4**

3b. Evergreen plants. . . . . . . . . . . . . . . . . . . . . . . . . . . . . . . . . . . . . . . . . . . . . . . . . . . . . . . . ...**7**

4a. Color of female flowers from pink to reddish . . . . . . . . . . . . . . . . . . . . . . . . . . . .***V. parviflora***

4b. Color of female flowers green, white, or yellow... . . . . . . . . . . . . . . . . . . . . . . . . . . . . . . . ..**5**

5a. Petiole length 50–60 cm . . . . . . . . . . . . . . . . . . . . . . . . . . . . . . . ... . . . . . . . . . . . .***V. carvalhoae***

5b. Petiole length 2–10 cm... . . . . . . . . . . . . . . . . . . . . . . . . . . . . . . . . . . . . . . . . . . . . . . . **6**

6a. Fruit length 2–8 cm, superior stamens filaments densely pubescent
. . . . . . . . . . . . . . . . . . . . . . . . . . . . . . . . . . . . . . . . . . . . . . . . . . . . . . . .***V. quercifolia***

6b. Fruit length 10–18 cm, superior stamens filaments glabrous or slightly pubescent
. . . . . . . . . . . . . . . . . . . . . . . . . . . . . . . . . . . . . . . . . . . . . . . . . ... . . . . . . .***V. candicans***

7a. Seed ellipsoidal-shaped and having pentalobed central leaflet. . . . . . . . . . .***V. pentalobis***

7b. Seed fusiform-shaped and lacking pentalobed central leaflet . . . . . . . . . . . . . . . . . ... . ..**8**

8a. Stigma having entire apex . . . . . . . . ... . . . . . . . . . . . . . . . . . . . . . . . . . . . . . . . ... . . . . .**9**

8b. Stigma having divided apex and fusiform seeds . . . . . . . . . ... . . . . . . . . . . . . . . . . ...**11**

9a. Seed surface having longitudinal ridges . . . . . . . . . . . . . . . . . . . . . . . . . . . . . . ...***V. microcarpa***

9b. Seed surface lacking longitudinal ridges . . . . . . . . . . . . . . . . . . . . . . . . . . . . . . . . . . ... . .**10**

10a. Plant completely covered by minute hairs and having ovoid-prolate fruits
. . . . . . . . . . . . . . . . . . . . . . . . . . ... . . . . . . . . . . . . . . . . . . . . . . . . . . . . . . .....***V. pubescens***

10b. Plants spindle-shaped and strongly angled fruits (fusiform) . . . . . . . ..***V. glandulosa***

11a. Seeds having smooth surfaces with rounded projections. . . . . . . ... . . ..***V. weberbaueri***

11b. Seeds having conical protuberances . . . . . . . . . . . . . . ... . . . . . . . . . . . . . . . . . . . . . . . . ...**12**

12a. Plants having rough-textured leaves and yellow-orange to pink fruits
. . . . . . . . . . . . . . . . . . . . . ... . . . . . . . . . . . . . . . . . . . . . . . . . . . . . . . . . . . . . ... . .***V. badilloi***

12b. Plants having glabrous leaves, inflorescence leaflets at its base, and ovoid fruits with acu-
minate apex . . . . . . . . . . . . . . . . . . . . . . . . . . . . . . . . . . . . . . . . . . . ...***V. chachapoyensis***

## Discussion

The assignment of accurate names for species is crucial, especially for those with confirmed agronomic potential as highland papayas. The taxonomy of these species, which are members of the genus *Vasconcellea*, has been mostly based on morphological characters and multilocus phylogeny, including detailed species descriptions and precise distribution maps [4, 5, 7, 45]. However, the use of additional methodologies and data sets is recommended to establish well-supported boundaries among species [17, 21]. Accordingly, six molecular markers have been used to delimit species in the genus *Vasconcellea* using phylogeny and four DNA-based methods. Although incongruence among some of these methods was observed in our analyses, genetic distance (ABGD, SPN), a coalescence method (BPP), and the multilocus phylogeny supported 22–25 different species in *Vasconcellea*, including five new species from northern Peru.

### Integrative approach

Our six loci phylogeny resulted in topology incongruence mainly to single loci phylogenies. However, multilocus sequence data are pivotal for the establishment of robust species delimitations [46, 47]. Incomplete lineage sorting, horizontal gene transfer, gene duplication and loss, hybridization, or recombination are probable explanations for this discordance [48]. Our data provided molecular evidence of hybridization, but natural processes such as introgression, chloroplast capture, selection, differentiation, mutations, and human selection might have all played a part generating evolving hybridizing species complexes in *Vasconcellea* [14, 20, 48, 49]. According to our multilocus phylogeny, 24 species (including 5 new species) were molecularly confirmed in *Vasconcellea* and are distributed in two main lineages, although previous studies grouped them into 3 clades [9, 44, 50]. One of these lineages, labelled clade 1 by d'Eeckenbrugge et al. [44], is composed of six taxa, including three new species, *V. badilloi*, *V. chachapoyensis*, and *V. pentalobis*. The restricted distribution of this lineage confirms its endemism to southern Ecuador and northern Peru with a high level of introgression history and sympatry [44]. The other lineage is composed of 18 species, including two new species, *V. carvalhoae* and *V. peruviensis*. This lineage contained clades 2 and 3 of d'Eeckenbrugge et al. [44], which are composed of specimens from different taxa but belong to sympatric populations with high morphological diversity related to hybrid segregation and phenotypic plasticity [14]. Strikingly, these two evolutionary lineages in *Vasconcellea* are molecularly well distinguished clades that can be considered two different genera. However, the lack of additional diagnostic features suggests that further analyses (e.g., anatomical observations on the basis of ultrastructure of vegetative and reproductive tissues and chemotaxonomic evaluations) must be accomplished before recognizing them as separate genera.

Regarding the genetic distance methods, similar results to those from the multilocus phylogeny were obtained by ABGD and SPN when delimiting *Vasconcellea* species. The additional

putative species identified with these methods mainly resulted from the split of *V. chacha-poyensis*, *V. pentalobis*, and *V. peruviensis* with the markers ITS, *psbA-trn*H, and *trnL-trn*F. This might suggest that these species encompass cryptic lineages [17] as a consequence of the initial hybridization process, but these splits are not supported by the multilocus phylogeny and BPP analyses dismissing crypticism.

In the coalescent methods, the presence of gene flow due to the high hybridization levels in different species of *Vasconcellea* had negative impacts, particularly on the GMYC model [51]. The GMYC model usually produces false positives and complex false positives when delimiting different taxa that have low or high magnitudes of gene flow, respectively [52, 53]. Nevertheless, the validation of BPP supports the status of the species recognized by the multilocus phylogeny (posterior probabilities, pp 0.61–0.99, S3 Table) and did not support those split or merged taxa by GMYC (pp lower than 0.29, S3 Table). Moreover, the additional species delimited by BPP is an unidentified *Vasconcellea* from Peru (IV17, IV18, IV20, IV21) which lacks of support to be considered a different entity. Therefore, sistership between this species and *V. stipulata* is not confirmed, suggesting that they might be conspecific. Additional specimens of this unidentified taxon should be sequenced to confirm its taxonomical status. The performance in empirical studies of the genetic and coalescent methods tends to undersplit and oversplit species, respectively [53–56]. However, our results suggest that ABGD and BPP are appropriate for determining diversity in *Vasconcellea* by recognizing those well-supported clades delimited by the multilocus phylogeny.

In our study, *V. pentalobis* and *V. peruviensis* were morphologically distinguished by their pentalobed central lobes and monoic inflorescence, respectively. Traits related to leaf (petiole length), female flowers (color, stigma shape), seeds (shape and texture), and fruits (length) slightly differentiated the other three new species. For instance, *V. badilloi* and *V. chacha-poyensis* have divided stigmas and conical protuberances on seed surfaces, while *V. carvalhoae* has an entire stigma and rounded projections on seed surfaces. The absence of robust morphological distinction traditionally occurs in organisms that lack complex structures such as fungi or algae, but the high phenotypic plasticity and hybridization scenarios in *Vasconcellea* might explain this absence in these plants [9, 44]. This morphological indistinctiveness among some *Vasconcellea* species was overcome by the application of molecular methods in plant taxonomy. In addition, our multilocus data, ABGD, and BPP analyses suggested conspecificity between *V. goudotiana*/*V. sphaerocarpa* as well as *V. pubescens*/*V. sprucei*. Therefore, *V. sphaerocarpa* (García-Barr. & Hern-Cam, 1958 in Badillo [7]), and *V. sprucei* (Badillo [57]) might be synonymized with *V. goudotiana* (Triana & Planch, 1873 in Badillo [7]) and *V. pubescens* (de Candolle [58]), respectively, on the basis of the principle of priority. However, further studies should delimit the relationships of those taxa including analyses of new material collected from type localities. Although several chloroplast and nuclear sequences have been used for assessing inter- and intraspecific relationships among species of Caricaceae [2, 9, 14, 20], only ITS and *trnL-trn*F intergenic showed better resolution for distinguishing species based on phylogeny and species delimitation methods. This suggests that initial screening regarding the diversity of *Vasconcellea* should include amplification of these markers. The segregation of five new species confirmed that phylogenetic diversity and DNA-species delimitation methods could be used to discover taxa within traditionally defined species [15, 17, 59].

## Conclusions

The use of an integrative approach to analyse diversity, including DNA-based delimitation methods, allowed the establishment of boundaries among species with morphological diversity, such as *Vasconcellea*, and thus provided support for the description of new taxa or

validated the taxonomic uncertainty of other *Vasconcellea* members. Our results demonstrated that the congruence among different methodologies applied in this integrative study (i.e., morphology, multilocus phylogeny, genetic distance, coalescence methods) are more likely to prove reliably supported species boundaries. Therefore, ABGD, BPP, and multilocus phylogeny are pivotal when establishing species boundaries in *Vasconcellea*.

## Supporting information

**S1 Fig. Phylogenetic tree based on maximum likelihood inference of combined *matK*, *psbA-trnH*, *rbcL*, *trnL-trnF* data.** Combination of markers were selected on the basis of high genetic pairwise divergence. Value above branches = Maximum likelihood bootstrap values (BS). The scale bar indicates the number of nucleotide substitution per site.
(JPG)

**S2 Fig. Phylogenetic tree based on maximum likelihood inference of combined *psbA-trnH*, *rbcL*, *trnL-trnF* data.** Combination of markers were selected on the basis of high genetic pairwise divergence. Value above branches = maximum likelihood bootstrap values (BS). The scale bar indicates the number of nucleotide substitutions per site.
(JPG)

**S3 Fig. Phylogenetic tree based on maximum likelihood inference of combined *matK*, *trnL-trnF* data.** Combination of markers were selected on the basis of high genetic pairwise divergence. Value above branches = maximum likelihood bootstrap values (BS). The scale bar indicates the number of nucleotide substitutions per site.
(JPG)

**S4 Fig. Phylogenetic tree based on maximum likelihood inference of combined *trnL-trnF* data.** Marker was selected on the basis of high genetic pairwise divergence. Value above branches = maximum likelihood bootstrap values (BS). Scale bar indicates the number of nucleotide substitutions per site.
(JPG)

**S5 Fig.** Bayesian inference ultrametric gene tree obtained using a prior Yule tree in BEAST with the statistical species delimitation results from GMYC based on ITS (A), *matK* (B) and *psbA-trnH* (C).
(JPG)

**S6 Fig.** Bayesian inference ultrametric gene tree obtained using a prior Yule tree in BEAST with the statistical species delimitation results from GMYC based on *rbcL* (A), *rpl20-rps12* (B) and *trnL-trnF* (C).
(JPG)

**S7 Fig.**
(JPG)

**S1 Table. List of primers used in the molecular analyses.**
(DOCX)

**S2 Table. Results of the Generalized Mixed Yule-Coalescent (GMYC) analyses under the single threshold model.**
(DOCX)

**S3 Table. Highest posterior probabilities of the six-gene Bayesian species delimitation analysis (BPP) by jointing species delimitation and species tree inference.**
(DOCX)

**S4 Table. Morphological comparisons among species of the genus *Vasconcellea*.**
(XLSX)

## Acknowledgments

We thank the editor and anonymous reviewers for comments that improved an earlier version of this manuscript.

## Author Contributions

**Conceptualization:** Daniel Tineo, Danilo E. Bustamante, Martha S. Calderon.

**Data curation:** Daniel Tineo, Eyner Huaman.

**Formal analysis:** Danilo E. Bustamante, Martha S. Calderon.

**Funding acquisition:** Manuel Oliva.

**Investigation:** Daniel Tineo, Danilo E. Bustamante, Martha S. Calderon, Jani E. Mendoza.

**Methodology:** Daniel Tineo, Danilo E. Bustamante, Martha S. Calderon, Jani E. Mendoza.

**Project administration:** Daniel Tineo, Eyner Huaman, Manuel Oliva.

**Resources:** Danilo E. Bustamante, Jani E. Mendoza, Eyner Huaman, Manuel Oliva.

**Software:** Danilo E. Bustamante.

**Supervision:** Danilo E. Bustamante, Martha S. Calderon.

**Validation:** Danilo E. Bustamante.

**Visualization:** Danilo E. Bustamante.

**Writing – original draft:** Daniel Tineo, Danilo E. Bustamante, Jani E. Mendoza, Eyner Huaman, Manuel Oliva.

**Writing – review & editing:** Danilo E. Bustamante, Martha S. Calderon.

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
