## [Decision Letter · Decision Letter 0]

29 Sep 2020

PONE-D-20-23252

High diversity of highland papayas (Caricaceae, Vasconcellea) from northern Peru revealed by an integrative approach includes five new species

PLOS ONE

Dear Dr. Bustamante,

Thank you for submitting your manuscript to PLOS ONE. After careful consideration, we feel that it has merit but does not fully meet PLOS ONE’s publication criteria as it currently stands. Therefore, we invite you to submit a revised version of the manuscript that addresses the points raised during the review process.

We look forward to receiving your revised manuscript.

Kind regards,

Himanshu Sharma

Academic Editor

PLOS ONE

Journal Requirements:

4.We note that [Figure(s) 1] in your submission contain [map/satellite] images which may be copyrighted. All PLOS content is published under the Creative Commons Attribution License (CC BY 4.0), which means that the manuscript, images, and Supporting Information files will be freely available online, and any third party is permitted to access, download, copy, distribute, and use these materials in any way, even commercially, with proper attribution. For these reasons, we cannot publish previously copyrighted maps or satellite images created using proprietary data, such as Google software (Google Maps, Street View, and Earth). For more information, see our copyright guidelines: http://journals.plos.org/plosone/s/licenses-and-copyright.

1.    You may seek permission from the original copyright holder of Figure(s) [1] to publish the content specifically under the CC BY 4.0 license. 

Additional Editor Comments (if provided):

Dear authors,

I have read through your work in detail and after getting reviewer's comments and I consider the current manuscript a valuable addition which enrich the existing genetic resources in Highland Papayas. The work is good and worth publishable and I am more than sure that if authors answer all the queries raised by reviewers it will be valuable addition in Papaya reserach. I have found a some major flaws in the manuscript that needs to be readdressed and MS should be majorly revised . In it's current form the paper cannot be considered for publication in Plos one.

Reviewers' comments:

Reviewer's Responses to Questions

**Comments to the Author**

1. Is the manuscript technically sound, and do the data support the conclusions?

Reviewer #1: Yes

Reviewer #2: Partly

Reviewer #3: Yes

Reviewer #4: Partly

2. Has the statistical analysis been performed appropriately and rigorously? 

Reviewer #1: Yes

Reviewer #2: Yes

Reviewer #3: Yes

Reviewer #4: Yes

3. Have the authors made all data underlying the findings in their manuscript fully available?

Reviewer #1: Yes

Reviewer #2: No

Reviewer #3: Yes

Reviewer #4: Yes

4. Is the manuscript presented in an intelligible fashion and written in standard English?

Reviewer #1: Yes

Reviewer #2: Yes

Reviewer #3: Yes

Reviewer #4: Yes

5. Review Comments to the Author

Reviewer #1: The paper looks sound. Provide a better title. Few queries that needs to be addressed are:

It does not reveal much about the integrated approach. Modify the review and discussion portion giving some background of 6 molecular markers, their application in phylogeny of other plants. For diversity analysis more number of individuals using other markers such as RAPD, ISSR, SSR, AFLP etc. are more suitable. Its rather a phylogeny study rather than diversity.

Reviewer #2: The present manuscript reports the molecular diversity in Papayas from northern Peru and concluded that high molecular diversity is prevailing in the analyzed germplasm. Language and overall structure of manuscript is good. They also proposed five new species based on their molecular data analyses/ works. As far as diversity is concerned paper contain good information on papayas, however, taxonomically I have many doubts which need to be clarified before the publication of this article my major issues are:

1. You are proposing five new species in this article, my question is: have you authenticated your data from any institutions/organization of your country which deals with the data related to taxonomic entities like any Botanical Garden or Botanical surveys Institutes or any other international organization like Kew Botanical Garden?

2. Comments or reports of these type of organizations are required to assign/ publish these new species as different species?

Other minor issues:

1. Title is not clear please revise it.

2. Abstract should contain clear cut statements about the work done, so please revise abstract.

3. The dendrogram/ phylogenetic tree given in figure is not clear/ words are not readable so please replace this figure by clear one.

4. You have mentioned morphological traits in your integrative approach but you did not show any analysis derived from morphological traits such as dendrogram etc.

Reviewer #3: Reviewers' comments:

The manuscript entitled " High diversity of highland papayas (Caricaceae, Vasconcellea) from northern Peru revealed by an integrative approach includes five new species" is an worthy effort, and it could be a very useful resource in area of plant science. However, it need to be address few comments before acceptance.

Major comments

Comment#1: Authors should rewrite the abstract; first six line of abstract should be 3-4 line, remaining authors should write his major findings.

Comment#2: Few place the sentences are too long, which are not clear for example line 44 to 48; 54 to 57, and 73 to 76; authors should split these to get reader easily understand.

Comment#3: The Figures 2 quality is not clear; however, I appreciate for remaining figures. It is really nice.

Comment#4: Reference section is not uniform, kindly flow uniformity, For example 43, 44, and 45 (In few doi is missing).

Reviewer #4: In the manuscript entitled as “High diversity of highland papayas (Caricaceae, Vasconcellea) from northern Peru revealed by an integrative approach includes five new species” authors have utilized integrated approaches to classify known species of papayas and ascertain new ones.

The overall quality of the manuscript is acceptable for publication in this journal. However, the identification of new species in the manuscript is mainly determined on the bases different molecular methods using different markers (ITS, matK, psbA-trnH, rbcL, rpl20-prs12, trnL-trnF) whereas morphological data remains orphan.

The markers used in this study are not the representative of the entire genome and have shown discordant patterns (authors also agrees line no. 219-222, 230-233 and 242) that raises the questions on the correct identification of new species. Further, new identified species are not cryptic species; hence, it is essential to establish relationship between morphological and molecular data present in this study or go for chloroplast genome sequencing to support the outcome of the study.

6. PLOS authors have the option to publish the peer review history of their article (what does this mean?). If published, this will include your full peer review and any attached files.

Reviewer #1: **Yes: **Abhishek Bhandawat

Reviewer #2: No

Reviewer #3: No

Reviewer #4: No

---

## [Author Response · Author response to Decision Letter 0]

8 Oct 2020

Dear Dr. Himanshu Sharma

Academic Editor

PLOS ONE

PONE-D-20-23252

"High diversity of highland papayas (Caricaceae, Vasconcellea) from northern Peru revealed by an integrative approach includes five new species"

Please, kindly find our response. We have accepted all suggestions and comments mentioned by editor and reviewers.

Editor's Comments to Authors

Journal Requirements:

FIXED: All additional requirements where included in the revised version of the manuscript.

FIXED: The permit for this research was added. This information was included in lines 110-112 in the material and methods section.

FIXED: We have included all accession numbers and Herbarium vouchers in Table 1 and 2 since it was recently released by NCBI and CHAX Herbarium, thus we have provided repository information for our data.

4. We note that [Figure(s) 1] in your submission contain [map/satellite] images which may be copyrighted. All PLOS content is published under the Creative Commons Attribution License (CC BY 4.0), which means that the manuscript, images, and Supporting Information files will be freely available online, and any third party is permitted to access, download, copy, distribute, and use these materials in any way, even commercially, with proper attribution. For these reasons, we cannot publish previously copyrighted maps or satellite images created using proprietary data, such as Google software (Google Maps, Street View, and Earth). For more information, see our copyright guidelines: http://journals.plos.org/plosone/s/licenses-and-copyright.

COMMENT: We have not used copyrighted [map/satellite] images. Our map was obtained from the Geoportal of the National Geographic Institute of Peru (IGN) in shapefile format with a DATUM WGS 1984. These are public domain. These information was added in Fig 1 caption in lines 127-129.

Additional Editor Comments: I have read through your work in detail and after getting reviewer's comments and I consider the current manuscript a valuable addition which enrich the existing genetic resources in Highland Papayas. The work is good and worth publishable and I am more than sure that if authors answer all the queries raised by reviewers it will be valuable addition in Papaya reserach. I have found a some major flaws in the manuscript that needs to be readdressed and MS should be majorly revised . In it's current form the paper cannot be considered for publication in Plos one.

COMMENT: We have positively answered and fixed all the queries raised by editor and reviewers.

Additional comments from Jazmin Toth:

3. In the Methods section, include a sub-section called "Nomenclature" 

FIXED: This information was included in lines 198-210. 

4. In the Results section, the globally unique identifier (GUID), currently in the form of a Life Science Identifier (LSID), should be listed under the new species name

FIXED: This information was included in lines 290, 341, 386, 452, 507. 

Reviewers' Comments to Authors

Reviewer #1: The paper looks sound. Few queries that needs to be addressed are: Provide a better title.

FIXED: The title was changed as follow: "An integrative approach reveals five new species of highland papayas (Caricaceae, Vasconcellea) from northern Peru".

It does not reveal much about the integrated approach. 

COMMENT: Integrated approach combines different methodologies in one effective system in order to generate stronger conclusions. Accordingly, our study used (i) morphology, (ii) phylogeny, (iii) genetic distance and (iv) coalescent delimitation methods to support species boundaries. Recent studies used the term "integrative approaches" when referring to those methodologies (Pavan and Marroig 2016, Sun et al. 2016, Surveswaran et al. 2018, Lee et al. 2019). Additionally, these references were included in line 98 of the revised version of the manuscript.

Modify the review and discussion portion giving some background of 6 molecular markers, their application in phylogeny of other plants. For diversity analysis more number of individuals using other markers such as RAPD, ISSR, SSR, AFLP etc. are more suitable. Its rather a phylogeny study rather than diversity.

COMMENT: A paragraph was included in lines 83-89 of introduction part giving some background for the 6 molecular markers and why RAPD, ISSR, SSR, AFLP are not recommended. Additionally, lines 672-675 were included in discussion part regarding the use of these molecular markers.

Reviewer #2: The present manuscript reports the molecular diversity in Papayas from northern Peru and concluded that high molecular diversity is prevailing in the analyzed germplasm. Language and overall structure of manuscript is good. They also proposed five new species based on their molecular data analyses/ works. As far as diversity is concerned paper contain good information on papayas, however, taxonomically I have many doubts which need to be clarified before the publication of this article my major issues are:

1. You are proposing five new species in this article, my question is: have you authenticated your data from any institutions/organization of your country which deals with the data related to taxonomic entities like any Botanical Garden or Botanical surveys Institutes or any other international organization like Kew Botanical Garden?

COMMENT: We have consulted several Herbaria located in Peru (Lima, Trujillo) looking for specimens of Vasconcellea but there have not been any records we can take a look or compare with. Our material has also been compared with specimens of Caricaceae that are deposited in the following virtual herbariums JSTOR Global Plants (https://plants.jstor.org/), the New York Botanical Garden Steere herbarium (http://sweetgum.nybg.org/science/), the Global Biodiversity Information Facility (https://www.gbif.org/), and Tropicos from Missouri Botanical Garden (http://www.tropicos.org). This information was included in lines 118-125 of the revised version of the manuscript.

2. Comments or reports of these type of organizations are required to assign/ publish these new species as different species?

COMMENT: The description of new taxa does not require comments or reports from these organizations. We have extensively showed evidence that the Peruvian taxa are new entities on the basis of (i) morphology, (ii) phylogeny, (iii) genetic distance and (iv) coalescent delimitation methods. 

Other minor issues:

1. Title is not clear please revise it.

FIXED: The title was changed as follow: "An integrative approach reveals five new species of highland papayas (Caricaceae, Vasconcellea) from northern Peru"

2. Abstract should contain clear cut statements about the work done, so please revise abstract.

FIXED: The abstract has been revised and rewritten following reviewer suggestions.

3. The dendrogram/ phylogenetic tree given in figure is not clear/ words are not readable so please replace this figure by clear one.

FIXED: The phylogram (Fig. 2) has been modified by enlarging the font size.

4. You have mentioned morphological traits in your integrative approach but you did not show any analysis derived from morphological traits such as dendrogram etc.

FIXED: A taxonomic key to distinguish among Peruvian species of Vasconcellea in lines 566-597 was added. Additionally, a supplementary material (S4 Table) and comments in lines 659-665 in discussion part containing comparison of morphological traits among all species of Vasconcellea were included.

Reviewer #3: Reviewers' comments:

The manuscript entitled " High diversity of highland papayas (Caricaceae, Vasconcellea) from northern Peru revealed by an integrative approach includes five new species" is an worthy effort, and it could be a very useful resource in area of plant science. However, it need to be address few comments before acceptance.

Major comments

Comment#1: Authors should rewrite the abstract; first six line of abstract should be 3-4 line, remaining authors should write his major findings.

FIXED: The abstract has been revised and rewritten following reviewer suggestions.

Comment#2: Few place the sentences are too long, which are not clear for example line 44 to 48; 54 to 57, and 73 to 76; authors should split these to get reader easily understand.

FIXED: Long sentences (lines 53 to 57; 63 to 66, and 89 to 93) were rewritten by splitting them to get reader easily understand.

Comment#3: The Figures 2 quality is not clear; however, I appreciate for remaining figures. It is really nice.

FIXED: The phylogram (Fig. 2) has been modified by enlarging the font size.

Comment#4: Reference section is not uniform, kindly flow uniformity, For example 43, 44, and 45 (In few doi is missing).

COMMENT. We checked the uniformity of references and some of them as 43 and 44 do not have a doi assigned, and abbreviations for the journals are not available. Reference 45 do have doi and is properly cited.

Reviewer #4: In the manuscript entitled as “High diversity of highland papayas (Caricaceae, Vasconcellea) from northern Peru revealed by an integrative approach includes five new species” authors have utilized integrated approaches to classify known species of papayas and ascertain new ones.

The overall quality of the manuscript is acceptable for publication in this journal. However, the identification of new species in the manuscript is mainly determined on the bases different molecular methods using different markers (ITS, matK, psbA-trnH, rbcL, rpl20-prs12, trnL-trnF) whereas morphological data remains orphan.

FIXED: A taxonomic key to distinguish among Peruvian species of Vasconcellea in lines 566-597 was added. Additionally, a supplementary material (S4 Table) and comments in lines 659-665 in discussion part containing comparison of morphological traits among all species of Vasconcellea were included.

The markers used in this study are not the representative of the entire genome and have shown discordant patterns (authors also agrees line no. 219-222, 230-233 and 242) that raises the questions on the correct identification of new species. 

COMMENT: Additional information was included in lines 81-89 of introduction part giving some background for the usefulness of the chloroplast (trnL–trnF, rpl20–rps12, psbA–trnH intergenic spacers, matK and rbcL genes) and nuclear (ITS) markers in delimiting species in Caricaceae by assessing inter- and intraspecific relationships among species of Caricaceae. These markers delimited species in a multilocus approaches instead of single locus.

Further, new identified species are not cryptic species; hence, it is essential to establish relationship between morphological and molecular data present in this study or go for chloroplast genome sequencing to support the outcome of the study.

COMMENT AND FIXED: In our study, we did not conclude crypticism among the new species. In fact, we have found some morphological differences that are summarized in the taxonomic key and supplementary S4 Table that support our molecular data. 

Thank you for your consideration,

Sincerely,

Ph.D. Danilo E. Bustamante

Universidad Nacional Toribio Rodríguez de Mendoza 

danilo.bustamante@untrm.edu.pe

---

## [Decision Letter · Decision Letter 1]

23 Oct 2020

PONE-D-20-23252R1

An integrative approach reveals five new species of highland papayas (Caricaceae, Vasconcellea) from northern Peru

PLOS ONE

Dear Dr. Danilo Bustamante 

Thank you for submitting your manuscript to PLOS ONE. After careful consideration, we feel that it has merit but does not fully meet PLOS ONE’s publication criteria as it currently stands. Therefore, we invite you to submit a revised version of the manuscript that addresses the points raised during the review process.

We look forward to receiving your revised manuscript.

Kind regards,

Himanshu Sharma

Academic Editor

PLOS ONE

Additional Editor Comments (if provided):

The manuscript entitled PONE-D-20-23252R1 An integrative approach reveals five new species of highland papayas (Caricaceae, Vasconcellea) from northern Peru has been extensively reviewed by all the reviewers and they agreed for the acceptance, but one reviewer has raised query that they have not provided the relationship of studied species on the basis of morphology in the form of deprogram which can provide additional evidence for the study and also some grammatical errors which needs to be corrected before the final approval.

Reviewers' comments:

Reviewer's Responses to Questions

**Comments to the Author**

1. If the authors have adequately addressed your comments raised in a previous round of review and you feel that this manuscript is now acceptable for publication, you may indicate that here to bypass the “Comments to the Author” section, enter your conflict of interest statement in the “Confidential to Editor” section, and submit your "Accept" recommendation.

Reviewer #1: All comments have been addressed

Reviewer #2: All comments have been addressed

Reviewer #3: All comments have been addressed

Reviewer #4: All comments have been addressed

2. Is the manuscript technically sound, and do the data support the conclusions?

Reviewer #1: Yes

Reviewer #2: Partly

Reviewer #3: Yes

Reviewer #4: Yes

3. Has the statistical analysis been performed appropriately and rigorously? 

Reviewer #1: Yes

Reviewer #2: Yes

Reviewer #3: Yes

Reviewer #4: Yes

4. Have the authors made all data underlying the findings in their manuscript fully available?

Reviewer #1: Yes

Reviewer #2: Yes

Reviewer #3: Yes

Reviewer #4: Yes

5. Is the manuscript presented in an intelligible fashion and written in standard English?

Reviewer #1: Yes

Reviewer #2: No

Reviewer #3: Yes

Reviewer #4: Yes

6. Review Comments to the Author

Reviewer #1: (No Response)

Reviewer #2: Although, authors have addressed all the raised issues but still they have not provided the relationship of studied species on the basis of morphology in the form of deprogram which can provide additional evidence for the study. I also advise authors to revise and thoroughly examine the language of manuscript to avoid ambiguity and to make your story more specific and objective oriented. For example in conclusion : starting of conclusion is not relevant neither it contain concrete information of your work. in line 678 what do you mean by others? please also check entire manuscript for such type of use of words and be specific. After fixing these issues manuscript can be accepted.

Reviewer #3: The manuscript entitled "An integrative approach reveals five new species of highland papayas (Caricaceae, Vasconcellea) from northern Peru" is now suitable for publication.

Reviewer #4: (No Response)

7. PLOS authors have the option to publish the peer review history of their article (what does this mean?). If published, this will include your full peer review and any attached files.

Reviewer #1: No

Reviewer #2: No

Reviewer #3: No

Reviewer #4: **Yes: **Pradeep Singh

---

## [Author Response · Author response to Decision Letter 1]

28 Oct 2020

Dear Dr. Himanshu Sharma

Academic Editor

PLOS ONE

PONE-D-20-23252R1

"An integrative approach reveals five new species of highland papayas (Caricaceae, Vasconcellea) from northern Peru"

Please, kindly find our response.

Editor's Comments to Authors

Additional Editor Comments: The manuscript entitled PONE-D-20-23252R1 An integrative approach reveals five new species of highland papayas (Caricaceae, Vasconcellea) from northern Peru has been extensively reviewed by all the reviewers and they agreed for the acceptance, but one reviewer has raised query that they have not provided the relationship of studied species on the basis of morphology in the form of deprogram which can provide additional evidence for the study and also some grammatical errors which needs to be corrected before the final approval.

COMMENTAND FIXED: In the previous review, we have included a table summarizing morphological features and a dichotomous key features among species of Vasconcellea as additional evidence for the morphological analyses since the dendrogram is not showing any additional evidence as you can notice in the attached figure (Fig. 1). This analyses poorly resolved several species as only one (e.g. V. sprucei, V. pulchra, V. candicans, V. glandulosa) and additionally it resulted in very low bootstrap support (blue numbers). We kindly ask you to reconsider the inclusion of this information in the manuscript.

Fig. 1. Dendrogram based on a hierarchical cluster analysis for 25 species of Vasconcellea. This dendrogram was drawn based on features Table S4 and a hierarchical cluster analysis using single linkage (nearest-neighbour) procedure using DARwin computer software version 6 for the 25 species of Vasconcellea.

In addition, grammatical errors pointed out by reviewer 2 have been fixed.

Reviewers' Comments to Authors

Reviewer #2: Although, authors have addressed all the raised issues but still they have not provided the relationship of studied species on the basis of morphology in the form of deprogram which can provide additional evidence for the study. I also advise authors to revise and thoroughly examine the language of manuscript to avoid ambiguity and to make your story more specific and objective oriented. 

COMMENT AND FIXED: In the previous review, we have included a table summarizing morphological features and a dichotomous key features among species of Vasconcellea as additional evidence for the morphological analyses since the dendrogram is not showing any additional evidence as you can notice in the attached figure (Fig. 1). This analyses poorly resolved several species as only one (e.g. V. sprucei, V. pulchra, V. candicans, V. glandulosa) and additionally it resulted in very low bootstrap support (blue numbers). We kindly ask you to reconsider the inclusion of this information in the manuscript.

Fig. 1. Dendrogram based on a hierarchical cluster analysis for 25 species of Vasconcellea. This dendrogram was drawn based on features Table S4 and a hierarchical cluster analysis using single linkage (nearest-neighbour) procedure using DARwin computer software version 6 for the 25 species of Vasconcellea.

For example in conclusion : starting of conclusion is not relevant neither it contain concrete information of your work. in line 678 what do you mean by others? please also check entire manuscript for such type of use of words and be specific. After fixing these issues manuscript can be accepted.

COMMENT AND FIXED: Grammatical errors pointed out by reviewer 2 have been fixed. Starting line of conclusion was removed and lines 58, 273, 590, 636, 654, 659, 673, and 678.

Thank you for your consideration,

Sincerely,

Ph.D. Danilo E. Bustamante

Universidad Nacional Toribio Rodríguez de Mendoza 

danilo.bustamante@untrm.edu.pe

---

## [Decision Letter · Decision Letter 2]

3 Nov 2020

An integrative approach reveals five new species of highland papayas (Caricaceae, Vasconcellea) from northern Peru

PONE-D-20-23252R2

Dear Dr. Bustamante,

We’re pleased to inform you that your manuscript has been judged scientifically suitable for publication and will be formally accepted for publication once it meets all outstanding technical requirements.

Kind regards,

Himanshu Sharma

Academic Editor

PLOS ONE

Additional Editor Comments (optional):

After second revision of the manuscript "An integrative approach reveals five new species of highland papayas (Caricaceae, Vasconcellea) from northern Peru" the reviewer is agreed to accept the manuscript. The authors have satisfactory answered the all queries. So the manuscript will be accepted for publication after correcting any mistakes which can be corrected at the time of proofreads.

Reviewers' comments:

Reviewer's Responses to Questions

**Comments to the Author**

1. If the authors have adequately addressed your comments raised in a previous round of review and you feel that this manuscript is now acceptable for publication, you may indicate that here to bypass the “Comments to the Author” section, enter your conflict of interest statement in the “Confidential to Editor” section, and submit your "Accept" recommendation.

Reviewer #2: All comments have been addressed

2. Is the manuscript technically sound, and do the data support the conclusions?

Reviewer #2: Yes

3. Has the statistical analysis been performed appropriately and rigorously? 

Reviewer #2: Yes

4. Have the authors made all data underlying the findings in their manuscript fully available?

Reviewer #2: Yes

5. Is the manuscript presented in an intelligible fashion and written in standard English?

Reviewer #2: Yes

6. Review Comments to the Author

Reviewer #2: The authors have revised the manuscript as per suggestions and have incorporated the changes where ever needed. It can be accepted in its present form.

7. PLOS authors have the option to publish the peer review history of their article (what does this mean?). If published, this will include your full peer review and any attached files.

Reviewer #2: No

---

## [Editor Report · Acceptance letter]

1 Dec 2020

PONE-D-20-23252R2 

An integrative approach reveals five new species of highland papayas (Caricaceae, Vasconcellea) from northern Peru 

Dear Dr. Bustamante:

I'm pleased to inform you that your manuscript has been deemed suitable for publication in PLOS ONE. Congratulations! Your manuscript is now with our production department. 

Kind regards, 

on behalf of

Dr. Himanshu Sharma 

Academic Editor

PLOS ONE